# Informativeness of teleconnections in frequency analysis of rainfall extremes

Andrea Magnini[1], Valentina Pavan[2], and Attilio Castellarin[1]

[1]Department of civil, environmental, chemical and materials engineering (DICAM), University of Bologna, Italy
[2]ARPAE-SIMC Emilia Romagna, Bologna, Italy

**Correspondence:** Andrea Magnini (andrea.magnini@unibo.it)

**Abstract.** We propose an effective and reproducible framework to assess the informative content of teleconnections (or climate indices) for representing and modeling the frequency regime of rainfall extremes at regional scale. Our dataset consists of 680 annual maximum series of rainfall depth, with 1 and 24 hours durations, located in Northern Italy. We compute at-site time series of L-moments (i.e., the mean and the L-coefficient of variation) through sliding time windows; then we discretize the study region into tiles, where L-moments time series are averaged. We compute the 30-years sliding mean for six teleconnections: North Atlantic Oscillation, Pacific Decadal Oscillation, East Atlantic – West Russia pattern (EA-WR), El Niño Southern Oscillation, Mediterranean Oscillation Index, and Western Mediterranean Oscillation Index (WeMOI). Then, we calculate L-moments-teleconnection Spearman correlations for single sites and for tiles with several resolutions, and retain correlations with p-values≤0.05. We observe spatial patterns of strong correlation between several teleconnections and gridded L-moments. These spatial patterns are clearly visible at various tiles' resolutions, and may be used for setting up regional prediction models. The strongest influence is detected for the sliding mean on the WeMOI and EA-WR. Finally, we show a preliminary application of climate-informed regional frequency analysis, through a hierarchical framework, where the L-moments are modelled as functions of teleconnections. We observe high variability of teleconnection-driven predictions of rainfall percentiles, and an increase in overall goodness-of-fit of the climate-informed regional models relative to stationary models. Overall, our research suggests promising pathways for climate-informed local and regional frequency analysis of rainfall extremes, and describes a general method, that can be adapted to different geographical and climatic contexts, as well as environmental variables.

## 1 Introduction

There is strong evidence that large-scale climate oscillations, also called climate indices or teleconnections, have a significant influence on a region's climate (e.g., Bardossy and Plate, 1992; Bonsal and Shabbar, 2008; Rasouli et al., 2020). Several authors have investigated the link between teleconnections and the seasonal regime of precipitation, and found strong influence for monthly (e.g., Das et al., 2020; Romano et al., 2022) or 3-month (i.e., seasonal, e.g., Belkhiri and Krakauer, 2023;

González-Pérez et al., 2022) cumulate rainfall and number of wet days (e.g., Ouachani et al., 2013; Ríos-Cornejo et al., 2015). Other authors focused on rainfall extremes, and showed how to exploit teleconnections to model the non-stationarity of the frequency regime of annual maxima (e.g., Cheng and AghaKouchak, 2014; Fauer and Rust, 2023; Ouarda et al., 2019; Ragno et al., 2018). Not only the parameters of the frequency distribution can be represented as a function of teleconnections (e.g., Johnson et al., 2025; El Adlouni and Ouarda, 2009), but also the distribution itself may change (Ouarda et al., 2019). Investigations on the balance between increased complexity and better reliability from non-stationary frequency analysis of rainfall extremes point out that the improvement can be worth the effort (Ouarda et al., 2020). Overall, the last findings appear to suggest the need for non-stationary frequency analysis of rainfall extremes, depending on teleconnections (e.g., Volpi et al., 2024; Nerantzaki and Papalexiou, 2022). Most of the studies on this topic show applications with a limited number of stations, where observations are abundant enough for fitting non-stationary local frequency analysis models. However, the dependence of the extreme rainfall regime on teleconnections may have regional patterns, instead of being specific for some isolated sites. In case this dependence could be described as a function of space, this would lead to non-stationary models for regional frequency analysis, similarly to the case of local frequency analysis.

The presence of regional structures in the dependence between teleconnections and rainfall has been investigated by several authors, focusing on monthly/seasonal or annual totals (Caroletti et al., 2021; Das et al., 2020; Ríos-Cornejo et al., 2015) or droughts (Romano et al., 2022). Differently, this field remains highly unexplored for rainfall extremes. This is a complex problem, as rainfall extremes have higher statistical and spatial complexity than seasonal/monthly rainfall totals or number of wet days. Thus, the correlation between teleconnections and extreme rainfall can vary significantly in sign, strength and significance across a region (see e.g., Jayaweera et al., 2023). In fact, climate may have strong local variations due to orography (Marra et al., 2021), which makes it difficult to understand which teleconnections are more relevant to a specific region. Moreover, the length and quality of the observed timeseries play an important role in the reliability of the obtained results (Martins and Stedinger, 2000; Nerantzaki and Papalexiou, 2022; Ouarda et al., 2020).

In this study, we propose a framework for assessing the informativeness of teleconnections in frequency analysis of hourly and daily rainfall extremes. In particular, we want to investigate (1) whether it is possible to delineate robust regional zonation of the dependence on teleconnections, and (2) what is the effect and suitability of teleconnection-informed regional frequency analysis. Accordingly, the study and proposed framework are structured in two parts: correlation analysis and regional frequency analysis. Our study area is North-Central Italy, where 680 timeseries with at least 30 years of records are available. We focus on annual maxima of precipitation with duration of 1 and 24 hours. In the first part of the research, we consider six teleconnection patterns with proven influence on the rainfall regime in the study area (Caroletti et al., 2021; Criado-Aldeanueva and Soto-Navarro, 2020; Krichak et al., 2014): North Atlantic Oscillation (NAO), Pacific Decadal Oscillation (PDO), East Atlantic – West Russia pattern (EA-WR), El Niño Southern Oscillation (ENSO), Mediterranean Oscillation Index (MOI), and Western Mediterranean Oscillation Index (WeMOI).

Differently from other studies, we do not perform our correlation analysis on the raw timeseries of the teleconnections and annual maxima. Here, two strategies are simultaneously adopted, that is to aggregate the data temporally and spatially. First, we consider sliding time windows, which allows us to (a) account only for long-term variability components, and (b) con-

sider the variation of the timeseries' statistics during the recorded period. In particular, we consider two linear moments (or L-moments, see Hosking and Wallis, 1997): the mean and L-coefficient of variation (L-CV) for each station. Second, we divide the study region into tiles: within each tile, we average the at-site sample L-moments, in order to obtain timeseries of regional L-moments. Finally, we compute tile-wise the correlation between the timeseries of the L-moments and the rolling mean of the teleconnections and we define raster maps of the dependence of the mean and L-CV. Significance testing specifically addresses autocorrelation issues due to the use of sliding windows (Lun et al., 2023).

In the second part, we fit polynomial relationships between the L-moments and the most influent teleconnections. By using a hierarchical approach (see e.g. Gabriele and Arnell, 1991; Castellarin et al., 2001), we define regional Generalized Extreme Value distributions (GEV, see Jenkinson, 1955) in a stationary and "doubly-stochastic" framework. The authors use the terminology "doubly-stochastic" for frequency models where the parameters depend on stochastic variables, as teleconnections (see Serinaldi and Kilsby, 2018). We test hypothesis that the GEV parameters of rainfall extremes depend on teleconnections (i.e., doubly-stochastic model) through ad-hoc Monte Carlo experiments that take spatial correlation into account (see e.g., Castellarin et al., 2024). Finally, the research is enriched by a critical discussion on the generality and reproducibility of the proposed methodology. It is shown that beside the choice of the study area, our methods are innovative and universally applicable.

## 2 Methodological framework

We propose an innovative and structured methodological framework for assessing the effectiveness of teleconnections-informed frequency analysis of rainfall extremes. The general methodology is structured into two phases. Some elements of the procedure, which include numerical parameters and specific functions, need to be adjusted according to the specific study case; for the sake of brevity, all of these elements will be referred to as "parameters". The general methodology is represented in Figure 1 and described in this Section, while the parametrization adopted for this study is detailed in Section 3.2.

### 2.1 Phase 1: correlation analysis

The first part aims at (1) evaluating the possible relation between teleconnection indices and the local climate indices (i.e., extreme rainfall statistics), (2) investigating the spatial structure of the correlation, and (3) producing maps of the correlation structure over the study area. These objectives are obtained through three fundamental steps which we describe below. The first step is the definition of two sliding time windows (STWs). One STW is used for the teleconnections, with width $w_{tel}$ years. Over this STW, the mean of the teleconnection $\mu_{tel}$ is computed. Another STW is used for the Annual Maximum Series (AMS), with width $w_{AMS}$ years. Over this STW, the at-site mean ($\mu$) and L-coefficient of variation (L-CV, see Hosking and Wallis, 1997) of the AMS of rainfall depths are computed (see Figure 1.a). Thus, for each gauging station ($st$) time series of the mean ($\boldsymbol{\mu_{st}}$) and L-CV ($\boldsymbol{L\text{-}CV_{st}}$) are obtained; these have length $n - w_{AMS} + 1$, where $n$ is the number of years of observations for the considered site:

$$\boldsymbol{\mu_{st}} = \left\{ \mu_{1,st}, \mu_{2,st}, ..., \mu_{n-w_{AMS}+1,st} \right\} \tag{1}$$

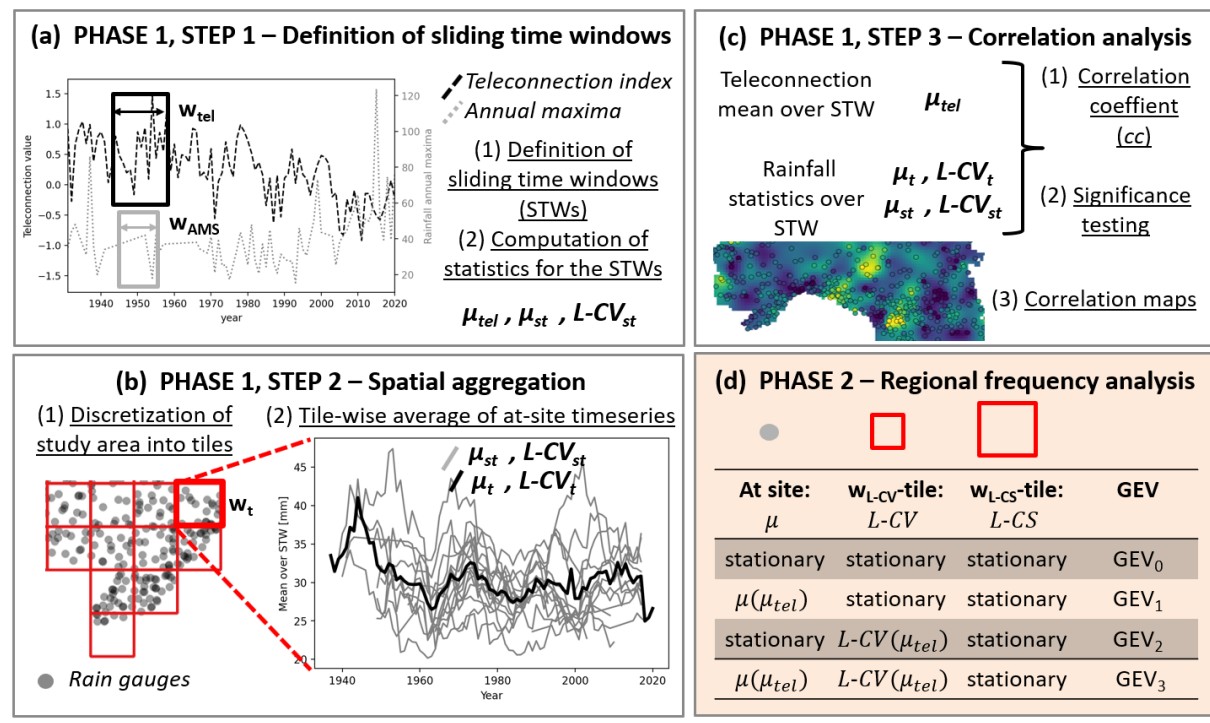

**Figure 1.** Methodological framework: first phase (white background, panels a, b and c) and second phase (coloured background, panel d). Vectors (i.e., time series) are highlighted with bold, italic font, substeps are numbered and underlined.

$$\boldsymbol{L\text{-}CV_{\text{st}}} = \{L-CV_{1,st}, L-CV_{2,st}, ...L-CV_{n-w_{AMS}+1,st}\} \tag{2}$$

where $\mu_{1,st}$, $\mu_{2,st}$ and $\mu_{n-w_{AMS}+1,st}$ represent the mean computed over the first, second and last time-steps defined by the STW at site $st$; the same notation is used for the L-CV. The value (i.e., mean or L-CV) computed at each time-step of the STW is conventionally assigned to the last year of the interval.

The second step is the discretization of the spatial domain into single tiles (or cells, or pixels) that do not overlap with each other. The spatial resolution is $w_t$. For each single tile ($t$), the timeseries $\boldsymbol{\mu_{st}}$ and $\boldsymbol{L\text{-}CV_{st}}$ of the gauged sites within the tile are

averaged yearly (see Figure 1.b). Thus, regional timeseries ($\boldsymbol{\mu_t}$ and $\boldsymbol{L\text{-}CV_t}$) for each tile are obtained:

$$\boldsymbol{\mu_t} = \left\{ \frac{\sum_{i=1}^{n_1} \mu_{1,i}}{n_1}, \frac{\sum_{i=1}^{n_2} \mu_{2,i}}{n_2}, ...., \frac{\sum_{i=1}^{n_{n-w_{AMS}+1}} \mu_{n-w_{AMS}+1,i}}{n_{n-w_{AMS}+1}} \right\} \tag{3}$$

$$\boldsymbol{L\text{-}CV_t} = \left\{ \frac{\sum_{i=1}^{n_1} L-CV_{1,i}}{n_1}, \frac{\sum_{i=1}^{n_2} L-CV_{2,i}}{n_2}, ..., \frac{\sum_{i=1}^{n_{n-w_{AMS}+1}} L-CV_{n-w_{AMS}+1,i}}{n_{n-w_{AMS}+1}} \right\} \tag{4}$$

where $n_1$, $n_2$, ...$n - w_{AMS} + 1$, are the numbers of stations with available rainfall statistics at steps 1,2, ..., $n - w_{AMS} + 1$ of the STW.

The third step is the correlation analysis (see Figure 1.c). The correlation coefficient ($cc$) is computed for each tile between the averaged timeseries ($\boldsymbol{\mu_t}$ and $\boldsymbol{L\text{-}CV_t}$) and the rolling mean of the considered teleconnection ($\boldsymbol{\mu_{tel}}$). We adopt the Spearman coefficient as it accounts also for non-linear correlations, yet preliminary experiments showed very similar results with the Pearson correlation coefficient. The significance of $cc$ is tested according to the methodology described in Lun et al. (2023), which takes into account the possible presence of spurious correlations, resulting from the use of sliding time windows. Only

stations with p-value$\leq 0.05$ are considered as significantly correlated (i.e., significance at 5%).

   Finally, we estimate the robustness, or reliability, of the detected correlation signals with an ad-hoc approach. This is an important step, since the definition of the parameters described above (i.e., $w_{tel}$, $w_{AMS}$, $w_t$) is necessarily affected by subjectivity and uncertainty. Then, the possible presence of spatial patterns of the correlation fields is investigated through the definition of correlation maps. The methodology adopted for correlation reliability assessment and correlation maps production is detailed

in Section 3.2.

## 2.2    Phase 2: regional frequency analysis

The second part of the present study aims at assessing (1) the possible effect of teleconnections on the frequency of extreme rainfall events, and (2) the potential of considering teleconnections as covariates in doubly-stochastic frequency models.

   For the sake of generality, the Generalized Extreme Value distribution (GEV, see Jenkinson, 1955) is considered, given its

flexibility and representativeness of the frequency regime of hydrological extremes (e.g., Papalexiou and Koutsoyiannis, 2013; Salinas et al., 2014). Nevertheless, the proposed framework is not limited to the case of the GEV distribution; on the contrary, it could be easily extended and adapted to the case in which alternative theoretical frequency distributions are considered and tested against each other. The cumulative distribution function of the GEV, $F_{GEV}(x)$, depending on the location, scale and shape parameters ($\xi, \alpha, k$), is defined as follows:

$$120 \quad F_{GEV}(x) = e^{-e^{-y}} \text{ , where } y = \begin{cases} -k^{-1} log_e[1 - k(x - \xi)/\alpha], & k \neq 0 \\ (x - \xi)/\alpha, & k = 0 \end{cases} \tag{5}$$

   The hierarchical method for regional frequency analysis is adopted (see e.g. Gabriele and Arnell, 1991). Accordingly, for a given gauged site, the mean is computed from the at-site records, the L-CV exploits all the time series within a tile with resolution $w_{L-CV}$ where the target site is included, and the L-CS exploits all the time series within a tile with resolution $w_{L-CS}$ (where $w_{L-CV} \leq w_{L-CS}$). The values adopted in the present study for $w_{L-CV}$ and $w_{L-CS}$ are detailed in Section

3.2.

   First, a stationary GEV is fit ($GEV_0$), where the mean over the whole time series is computed at-site, and the regional L-CV and L-CS are obtained as described in Hosking and Wallis (1997) as a weighted average over their respective tiles by referring to the complete sequences of annual maxima (see Figure 1).

Second, GEV distributions whose parameters depend on teleconnections are set up. Following the suggestion by Serinaldi and Kilsby (2018), we term these distributions as doubly-stochastic (DS) models, even if we acknowledge that the most common definition in the literature is "non-stationary" models (e.g., Volpi et al., 2024, and references therein). We consider three types of doubly-stochastic models. The first type, $GEV_1$, adopts the same L-CV and L-CS as the $GEV_0$, whereas the mean varies as a function of the best teleconnection index (selected in phase 1). This function is fitted at-site, according to the hierarchical regionalization framework. The choice for its shape, $f(x)$, is detailed in Section 3.2. The second type, $GEV_2$, adopts the same mean and L-CS as the $GEV_0$, while the L-CV varies as a function $f(x)$ of the best index. This is fitted on the $w_{L-CV}$-averaged timeseries ($\textbf{L-CV}_\textbf{t}$), according to the hierarchical regionalization framework. The third type, $GEV_3$, adopts the same L-CS as the $GEV_0$, while the mean and L-CV are obtained with the same methods as for the $GEV_1$ and $GEV_2$, respectively. Finally, the models are compared by means of the ratio of models' likelihood (RML). This metric is usually defined as "ratio of maximum likelihood", and is very commonly used for this purpose (e.g., Ashkar and Aucoin, 2012; Ashkar and Ba, 2017). It is defined as in Ashkar and Ba (2017):

$$RML = log_e\left(\frac{LH_{GEV_{DS}}}{LH_{GEV_0}}\right) \tag{6}$$

Where $LH_{GEV_0}$ and $LH_{GEV_{DS}}$ are the likelihood of the observed timeseries computed with the stationary and doubly-stochastic models, respectively.

Since RML is asymptotically distributed as half of a Chi-squared variable with degrees of freedom equal to the difference of the number of parameters of the two models (e.g., see Bhattacharya and Burman, 2016; Coles, 2001), a lower threshold $th_{RML}$ should be considered to conclude that $GEV_{DS}$ is a better fit to the data. Given that significance is tested at 5%, the theoretical significance threshold equals half of the 95th percentile of the corrispondent Chi-squared distribution. In the present study, $th_{RML}$ values are evaluated with reference to the total number of parameters of the models, consisting of those of the frequency distribution and those of $f(x)$. Furthermore, since in our opinion these two sets of parameters do not have the same weight (see also what observed by Laio et al., 2009), we have also determined $th_{RML}$ values experimentally, through a set of Monte Carlo (MC) experiments that are described in Appendix A. The thresholds obtained with the two methods, which depend on the choice of $f(x)$ are shown in Section 3.2.

Finally, we test field significance of possible doubly-stochastic spatial signals in presence of cross-correlation. In fact, the number of sites with doubly-stochastic rainfall regime could be inflated by the presence of intersite dependence (see also Castellarin, 2007; Vogel et al., 2001). For this reason, we define a significance test ad-hoc, based on the comparison of $n_{DS}$, the global number of sites where a given doubly-stochastic model outperforms the stationary model, with lower thresholds $th_{nDS}$. These thresholds are estimated through MC experiments that are described in Appendix B.

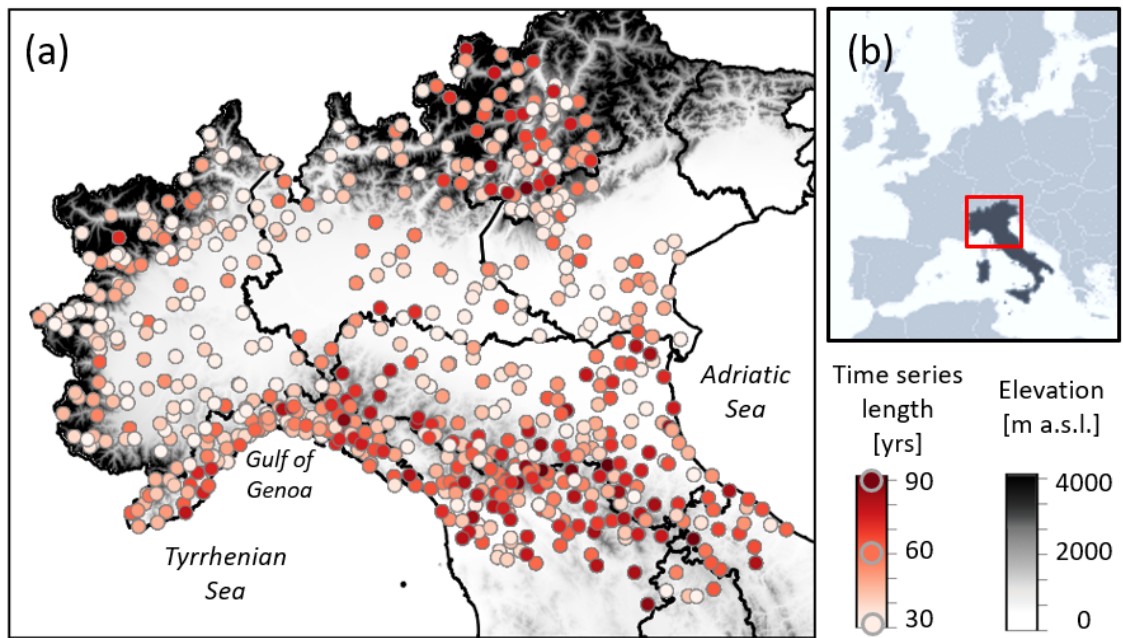

**Figure 2.** Elevation (a, in grey color scale) and location (b) of the study area. Length of the timeseries of sub-daily annual maximum rainfall depths (a, red color scale). Black lines: Italian administrative regions.

## 3 Study region and parameterization of the procedure

### 3.1 Study region and data

The study area includes most of Northern and part of Central Italy, a region characterized by great climate variability (see Figure 2.a). Two main mountain ranges are present: the Alps in the North, with a maximum elevation of about 4000 m a.s.l., and the Apennines, crossing all along continental Italy, with a maximum elevation of $\sim$2100 m a.s.l. in the study area. The largest Italian plain, the Po plain, is located at the southern border of the Alps, following the course of the Po River from the Northwest to the Northeast, where low coasts are located.

We select 680 gauged stations (Figure 2.a) from the I2-RED dataset (Mazzoglio et al., 2020) with a minimum of 30 years of data. Thus, all the selected timeseries should be sufficiently long to show variations of the frequency regime of rainfall extremes, if these are present (see also Renard et al., 2008; Ouarda et al., 2019). For each station, we consider time series of annual maximum cumulative rainfall over 1 and 24 consecutive hours, which represent distinct events: mainly convective the former, and mainly synoptic the latter. Data have been recorded between 1921 and 2020.

In the study, six teleconnections are considered; a detailed description of their nature is not reported here, since an interested reader can refer to the extensive literature cited in the text. Namely, these are the North Atlantic Oscillation (NAO, see Jones et al., 1997), the East Atlantic-West Russia (EA-WR) pattern (see Barnston and Livezey, 1987), the Pacific Decadal Oscillation

(PDO, see Zhang et al., 1997), El Nino Southern Oscillation (ENSO, see Chen et al., 2019), Mediterranean Oscillation Index (MOI, see Conte et al., 1991), and Western Mediterranean Oscillation Index (WeMOI, see Martin-Vide and Lopez-Bustins, 2006). All these indices have significant influences on local climate at several locations in Europe and the Mediterranean (e.g., Caroletti et al., 2021; Krichak et al., 2014, 2002; Krichak and Alpert, 2005). The NAO, EA-WR, PDO and ENSO are freely accessible from the NOAA Physical Sciences Laboratory data base available at https://psl.noaa.gov/data/climateindices/list/. The MOI and WeMOI can be retrieved from the University of East Anglia's Climate Research Unit (CRU; https://crudata.uea. ac.uk/cru/data/moi/).

## 3.2 Parametrization of the procedure for the study area

The parameterization of the methodology for the study area presented below results from several preliminary experiments. The implementation of the first part of the study requires the definition of $w_{tel}$, $w_{AMS}$, $w_t$, and the approach for the robustness analysis (i.e., the evaluation of the reliability of the correlation coefficients computed with $w_{tel}$, $w_{AMS}$ and $w_t$; see Section 2.1). First, $w_{tel}$ is set to 30 years. Preliminary experiments with smaller $w_{tel}$ provided similar results, yet 30 years generally smooth out short interannual oscillations, while preserving the pluridecadal climate variability. Also, shorter oscillations (i.e., <30 years) would not be of interest for the design of hydraulic structures, while longer ones (i.e., >30 years) would be highly uncertain to detect, due to limited length of the timeseries.

Second, $w_{AMS}$ is set to 10yrs. This choice is also a trade off between a minimum width for the computation of the L-CV and a minimum length of the timeseries of the rainfall statistics (i.e., $\boldsymbol{\mu}_{st}$ and $\boldsymbol{L\text{-}CV}_{st}$). In fact, on the one hand $w_{AMS}$ <10 years would lead to large sampling variability of local L-CV values. On the other hand, the $\boldsymbol{\mu}_{\mathbf{st}}$ and $\boldsymbol{L\text{-}CV}_{\mathbf{st}}$ timeseries originated from an n-long annual maxima timeseries have length of $n - (w_{AMS} - 1)$. This means that longer $w_{AMS}$ would lead to shorter $\boldsymbol{\mu}_{\mathbf{st}}$ and $\boldsymbol{L\text{-}CV}_{\mathbf{st}}$, which in turn would lead to a smaller number of stations where the correlation between teleconnections and rainfall statistics can be reliable.

Concerning the spatial resolution of the tile size, $w_t$, we consider four values: 0km (i.e., considering the single gauged stations, with no spatial discretization), 15km, 30km and 50km. This multiple choice comes from a balance. On the one hand, L-CV computed over a 10yrs time window may have low accuracy, which can be addressed by averaging L-CVs from several stations within large tiles (i.e., regional prediction). On the other hand, larger tiles may be less statistically homogeneous. Moreover, averaging L-statistics over large tiles may smooth the variability of the rainfall regime, hiding local patterns where the morphology is complex. Since there is no universal guideline for finding the right trade off, we decide to consider four different values for $w_t$. The suitability of these values is tested by means of the Hosking and Wallis (1997) heterogeneity measure for L-CV. The results of the test (not reported here for the sake of brevity) detected "definitely heterogeneous" tiles in a very limited number of cases for all the resolutions, confirming the viability of the homogeneity hypothesis.

We assess the robustness of the correlation signal at each station $st$ through a reliability index, $ri_{st}$, defined as:

$$ri_{st} = sign(cc_{0km,st}) + sign(cc_{15km,st}) + sign(cc_{30km,st}) + sign(cc_{50km,st}) \tag{7}$$

Where $cc_{0km,st}$ is the correlation computed at station $st$, while $cc_{15km,st}$, $cc_{30km,st}$ and $cc_{50km,st}$ are the correlation coefficients relative to the tiles (with $w_t$ 15km, 30km and 50km) where $st$ is inserted. Non-significant correlations are considered as 0. This is considered to be a measure of the spatial coherence of the correlation signal, which varies between -4 and 4. The absolute value is the coherence of the correlations at different tiles. The sign represents the sign of the prevailing correlation. For instance, a gauged station with $ri$ 4 indicates a significant and positive correlation when calculated at-site, as well as for 15km, 30km, and 50km aggregation. On the opposite, 0 represents areas with no significant correlation or where positive and negative correlations compensate with each other (e.g. positive correlation at-site and at 15km resolution, and negative correlation at 30km and 50km resolutions). Finally, $ri$ is interpolated by ordinary kriging (Hengl, 2007) to produce reliable maps of the correlation field of the mean and L-CV for 1h and 24h.

Regarding the second part of the study, $w_{L-CV}$, $w_{L-CS}$ and $f(x)$ need to be set. The selected resolution $w_{L-CV}$ is 30km, as it is a good trade off between accuracy of the prediction (i.e., aggregating at least two at-site L-CV timeseries) and an adequate representation of regional patterns and local variability. We set $w_{L-CS}$ to 100km, based on low spatial variability of the skewness parameter (e.g., Gabriele and Arnell, 1991; Claps et al., 2022) and improved accuracy.

The function $f(x)$ between teleconnections and L-statistics is shaped as a second-order polynomial function. This form has been selected because of its simplicity (i.e., only three parameters are needed) and adaptability to the empirical data, given that the observed dependence of extreme rainfall statistics on teleconnections is often non-linear. Even though other choices are possible, the aims of the present study are mainly demonstrative of the potential of the proposed approach. Thus, the nature and selection of the best function is not part of the main focus of our analyses.

The lower thresholds $th_{RML}$ considered for RML significance test at 5% (see Section 2.2) are reported in Table 1. Interestingly, our Monte Carlo experiments (see details in Appendix A) suggest that RML's diagnostic power is not affected by the duration and teleconnection considered. Also, the resulting empirical $th_{RML}$ values are much lower than the theoretical ones, and less affected by the number of parameters of $f(x)$ (compare $th_{RML}$ for $GEV_3$ with $GEV_1$ and $GEV_2$). While important for the general field of model evaluation, a detailed discussion of these results is beyond the scope of the present study. However, they present an interesting topic for further research.

|  | Number of parameters | Empirical $th_{RML}$ | Theoretical $th_{RML}$ |
|---|---|---|---|
| $GEV_1$ | 2+3 | 1.5 | 3 |
| $GEV_2$ | 2+3 | 1.5 | 3 |
| $GEV_3$ | 1+3+3 | 1.6 | 4.7 |

**Table 1.** RML significance thresholds, $th_{RML}$, at 5%

## 4 Results

 ### 4.1 Phase 1: correlation analysis

Table 2 reports the number of gauges (at-site), or tiles (resolutions from 15km to 30km), showing statistically significant correlations at 5% for each teleconnection and the two statistical moments considered here (i.e., the mean, $\mu$, and the L-coefficient of variation, L-CV) for durations of 1 and 24 hours.

| | 1h | | | | 24h | | | |
|---|---|---|---|---|---|---|---|---|
| | at-site | 15km | 30km | 50km | at-site | 15km | 30km | 50km |
| **NAO - $\mu$** | 118 | 61 | 25 | 7 | **110** | 55 | 19 | 9 |
| **EA-WR - $\mu$** | **131** | **97** | **53** | **32** | 106 | **58** | **30** | 10 |
| **PDO - $\mu$** | **123** | 62 | 28 | **12** | 101 | **53** | 22 | **5** |
| **ENSO - $\mu$** | 8 | 7 | 2 | 0 | 3 | 2 | 1 | 0 |
| **WeMOI - $\mu$** | **139** | **87** | **30** | **16** | **133** | 78 | 35 | **14** |
| **MOI - $\mu$** | 102 | 55 | 25 | 8 | 102 | 51 | 17 | 7 |
| **NAO - L-CV** | 96 | 48 | 21 | 10 | **101** | 49 | **27** | 10 |
| **EA-WR - L-CV** | 86 | 55 | **22** | 7 | 94 | 52 | **22** | 8 |
| **PDO - L-CV** | **101** | **57** | 20 | **11** | **102** | 61 | 24 | **9** |
| **ENSO - L-CV** | 8 | 4 | 3 | 2 | 4 | 3 | 2 | 0 |
| **WeMOI - L-CV** | 82 | 47 | **25** | 5 | **98** | 48 | **17** | 7 |
| **MOI - L-CV** | **107** | **59** | **22** | **11** | **98** | **55** | **25** | **11** |

**Table 2.** Number of significant correlations for all the considered teleconnections and tiles' resolutions. The numbers stand for stations in columns "at-site" and for tiles in the other columns. For each case (i.e., each semi-column), the highest number is marked with underlined bold font, while the second highest has bold font.

The most correlated indices are WeMOI and the EA-WR for $\mu$, and PDO and MOI for L-CV. ENSO shows the weakest
 influence on both statistics, and therefore is excluded from further analysis. The overall number of significant correlations is not sufficient to fully understand the dependence relationships of $\mu$-teleconnection or L-CV-teleconnection, nor to compare the influence of various teleconnections. Instead, we main objective is the description of the spatial correlation field, that is the spatial patterns of significant correlations, and their robustness to varying the resolution of the spatial aggregation (i.e., tiles' size). This is represented through the interpolated reliability index, $ri$, in Figure 3. As defined in Sections 2.1 and 3.2, areas
 where $ri \geq 3$ or $ri \leq -3$ indicate "consistent" correlation. Focusing on $\mu$, it is confirmed that WeMOI and EA-WR have a strong influence, particularly concentrated in two major patterns of consistent negative correlation in the Gulf of Genoa and in the North-Eastern Alps (panels b, d, g and i of Figure 3). These patterns are evident in the case of 1h duration, where the North-East shows consistent correlation with all the indices (panels a-e of Figure 3). For the 24h duration the two patterns for WeMOI and EA-WR are more fragmented and less extended, even tough still present, while the other indices show only small

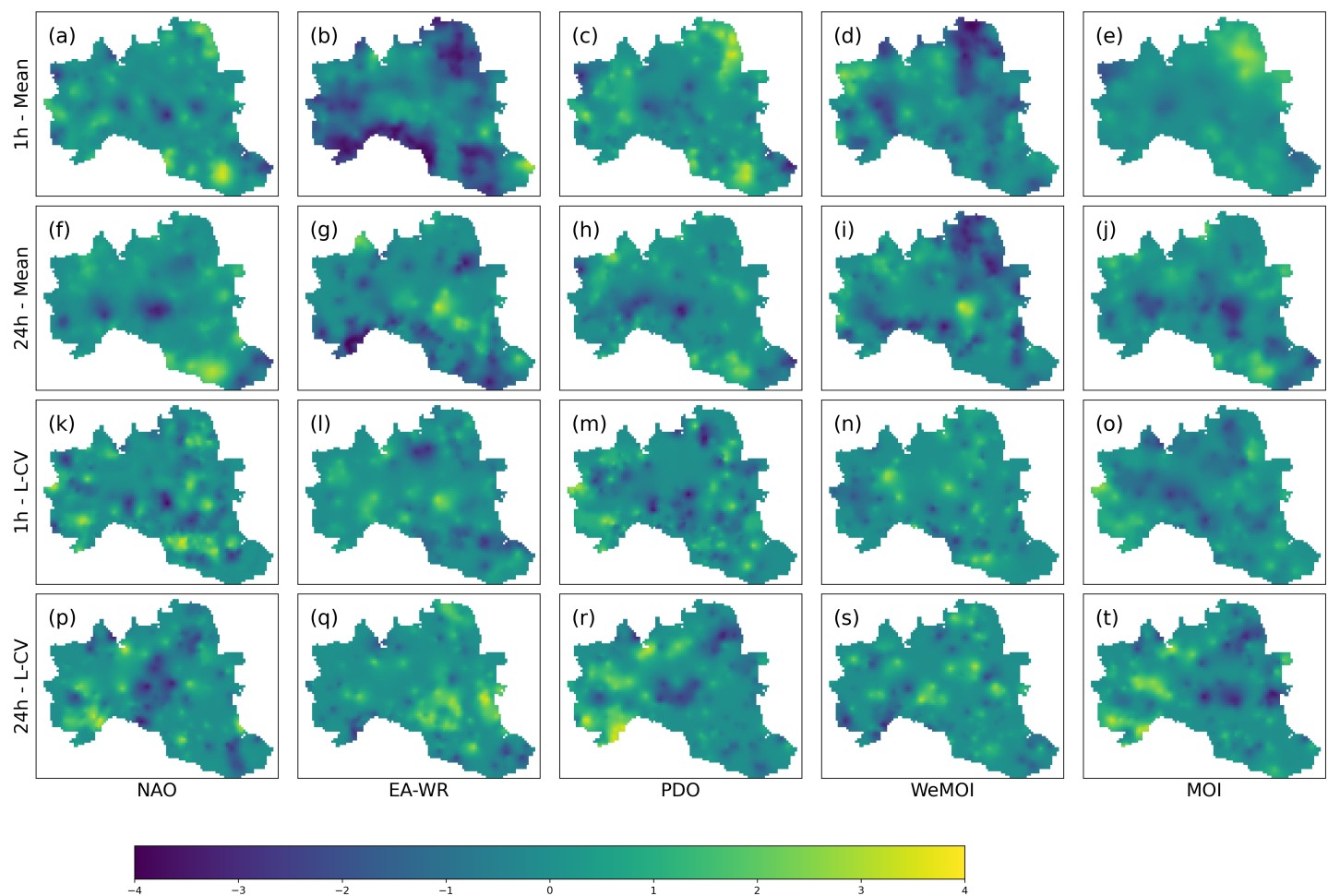

**Figure 3.** Raster maps of reliability index ($ri$) of the correlation between teleconnections and mean (panels a-j), or L-CV (panels k-t) of AMS of rainfall depths with duration of 1h (panels a-e and k-o) and 24h (panels f-j and p-t)

areas with strong influence on $\mu$.

Focusing on L-CV, consistent correlation patterns for 1h are highly heterogeneous, suggesting that regional $GEV_2$ models could be ineffective (panels k-o of Figure 3). More extended, yet still restrained, patterns can be observed for 24h, in particular in the western portion of the Gulf of Genoa and in the central region of the study area (see panels p-t in Figure 3). It is difficult to establish which index is the most influent, but PDO and MOI show high similarities and promising homogeneous patterns, 250     in agreement with Table 2.

The results obtained for WeMOI are presented in detail in Figure 4, which reports statistically significant Spearman correlation coefficients at different spatial resolutions for $\mu$ (panels a-h) and L-CV (panels i-p), respectively. Several stations present statistically significant correlation values with WeMOI, with signs and amplitude changing depending on the site considered

(panels a and e for $\mu$, i and m for L-CV, Figure 4). Aggregating stations into tiles reduces spatial heterogeneity and decreases the extension of significantly correlated areas, which allows to describe the geographical pattern of the correlation field (see other panels). Both for 1h and 24h duration, the correlation fields of extreme rainfall with $\mu$ present consistent spatial patterns, with extended areas characterized by homogeneous values (panels a-h). On the contrary, the correlation field of L-CV is more sensitive to the change of tiles' resolution and presents smaller isolated hotspots (panels i-p). Similar results are observed also for the other teleconnections, but they are not reported for the sake of brevity.

## 4.2 Phase 2: Frequency analysis

Following the adopted framework (see Sections 2.2 and 3.2), polynomial relations are fitted at-site (i.e., $\mu$-teleconnection) and at 30km-tiles (i.e., L-CV-teleconnection) to obtain the parameters for $GEV_1$ and $GEV_2$ distributions, respectively, and jointly for $GEV_3$. Then, the goodness-of-fit is compared with the stationary framework (i.e., $GEV_0$) with RML (equation 6). For the sake of brevity, the gauged sites where RML is higher than a given RML threshold (Table 1) will be hereinafter referred to as "doubly-stochastic sites". The number of doubly-stochastic sites in each case, $n_{DS}$, is reported in Table 3, while Figure 5 represents their location.

| | $n_{DS}$, RML $\geq$ empirical $th_{RML}$ | | | | | $n_{DS}$, RML $\geq$ theoretical $th_{RML}$ | | | | |
| --- | --- | --- | --- | --- | --- | --- | --- | --- | --- | --- |
| | **NAO** | **EA-WR** | **PDO** | **WeMOI** | **MOI** | **NAO** | **EA-WR** | **PDO** | **WeMOI** | **MOI** |
| **1h - $GEV_1$** | 101 | 105 | 113 | **171** | 85 | 31 | 45 | 33 | **61** | 22 |
| **24h - $GEV_1$** | 67 | 98 | 72 | **141** | 86 | 19 | 43 | 17 | **53** | 16 |
| **1h - $GEV_2$** | 58 | 51 | 60 | **85** | 46 | 12 | 16 | 12 | **16** | 10 |
| **24h - $GEV_2$** | 65 | 70 | 60 | **93** | 59 | 17 | 18 | **22** | 21 | 18 |
| **1h - $GEV_3$** | 152 | 173 | 169 | **248** | 149 | 22 | 36 | 26 | **50** | 18 |
| **24h - $GEV_3$** | 125 | 170 | 138 | **213** | 119 | 16 | 29 | 18 | **43** | 13 |

**Table 3.** Number of doubly-stochastic sites ($n_{DS}$). I.e., stations where RML $\geq th_{RML}$. The highest number is marked with bold font in each line, corresponding to a specific duration and doubly stochastic framework. Emprical and theoretical threshold are reported in Table 1

Regarding $GEV_1$, WeMOI has the highest number of doubly-stochastic sites for 1h, followed by EA-WR and PDO, depending on the threshold for RML (3 or 1.5, respectively, see Table 3). For 24h, WeMOI and EA-WR are confirmed as the most influent indices. Figure 5 is quite in agreement with Figure 3, confirming that in the two regions of the Gulf of Genoa and North-East $GEV_1$ fits the AMS better than $GEV_0$. These two regions show non-stationarity signals with all the indices (see panels a-j), but particularly with the most influent ones (panels b, d, g, i).

Regarding $GEV_2$, WeMOI has the highest number of doubly-stochastic sites both for 1h and 24h, followed closely by the other indices (see Table 3). In this case, detecting clear geographical patterns is difficult, but it is possible to spot some sub-regions where the signal is strong for multiple indices. Some examples are the North-East for 1h (see panels k-o of Figure 5), and the western portion of the Gulf of Genoa (in agreement with Figure 3) and the south-central portion of the Adriatic coast (see

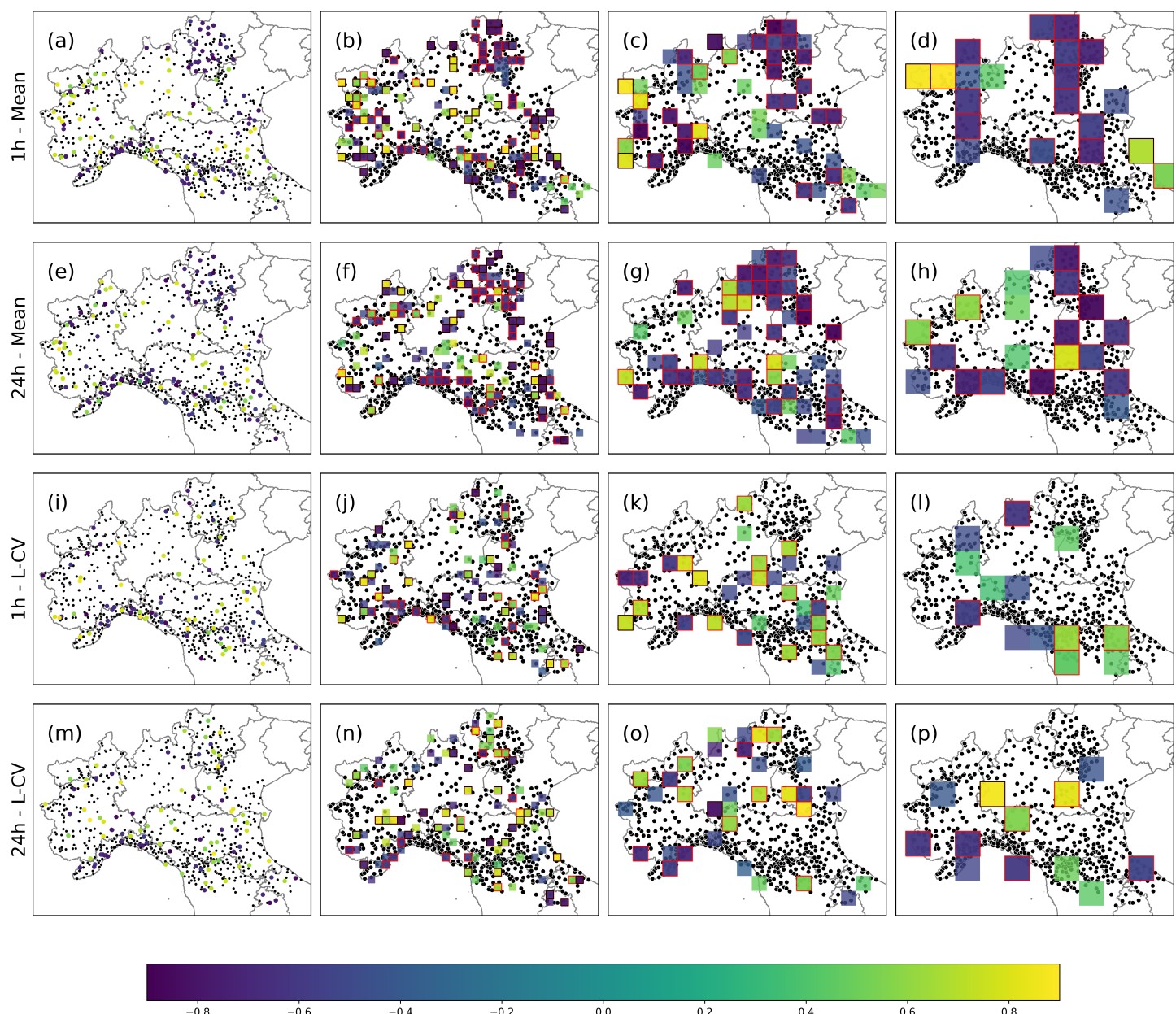

**Figure 4.** Spearman correlation coefficient between WeMOI and the mean [L-CV] for AMS of rainfall depths with duration of 1h and 24h: at-site (a [i] and e [m]) and for tiles of size 15km (b [j] and f [n]), 30km (c [k] and g [o]), and 50km (d [l] and h [p]). Statistically significant (at 10%) correlation coefficients are illustrated using a purple-green-yellow color scale. Red outlines highlight correlation significant at 5% (i.e., the ones considered for the $ri$). Black outlines highlight tiles where only one station is present. In panels a and i, gray circles represent non-significantly correlated stations

panels p-t of Figure 5).

Considering the theoretical threshold (i.e., 4.7) doubly-stochastic sites for $GEV_3$ have a very similar spatial configuration relative to $GEV_1$ or $GEV_2$, according to the case (see panels u-d1 of Figure 5). Hence, the numbers of $GEV_3$ doubly-stochastic sites in Table 3 are close to the maximum value within $GEV_1$ and $GEV_2$ for each index and duration. Differently, when the empirical threshold is considered (i.e., 1.6), the results suggest that $GEV_3$ doubly-stochastic sites are much more than $GEV_1$ and $GEV_2$, and located over the whole study area.

As mentioned in Section 2.2, the field significance of non-stationary (i.e., doubly-stochastic in this context) signals in presence of intersite dependence is tested by means of lower thresholds $th_{nDS}$ to the number of doubly-stochastic sites $n_{DS}$ (see details of the test in Appendix B). While WeMOI and EA-WR are clearly very effective for $GEV_1$ and $GEV_3$ distributions, it is hard to unequivocally establish the best teleconnections for $GEV_2$. For coherence, the test in the present study is performed for WeMOI and EA-WR for all of the three models, and the results are reported in Table 4. Regarding $GEV_1$, the number of doubly-stochastic sites is not significant when considering 1.5 as $th_{RML}$ (with the exception of WeMOI for 1h), while the null hypothesis (i.e., the distribution is stationary) is always rejected with a threshold of 3. Differently, for $GEV_2$ and $GEV_3$ the null hypothesis is rejected with all the thresholds.

|  | Empirical $th_{RML}$ | | Theoretical $th_{RML}$ | |
| --- | --- | --- | --- | --- |
|  | $n_{DS}$, RML $\geq$1.5 | $th_{nDS}$, RML $\geq$1.5 | $n_{DS}$, RML $\geq$3 | $th_{nDS}$, RML $\geq$3 |
| **EA-WR, 1h -** $GEV_1$ | 105 | **117** | **45** | 30 |
| **WeMOI, 1h -** $GEV_1$ | **171** | 139 | **61** | 41 |
| **EA-WR, 24h -** $GEV_1$ | 98 | **126** | **43** | 34 |
| **WeMOI, 24h -** $GEV_1$ | 141 | **148** | **53** | 47 |
| **EA-WR, 1h -** $GEV_2$ | **56** | 19 | **11** | 6 |
| **WeMOI, 1h -** $GEV_2$ | **85** | 24 | **16** | 8 |
| **EA-WR, 24h -** $GEV_2$ | **70** | 28 | **18** | 9 |
| **WeMOI, 24h -** $GEV_2$ | **93** | 31 | **36** | 11 |
|  | $n_{DS}$, RML $\geq$1.6 | $th_{nDS}$, RML $\geq$1.6 | $n_{DS}$, RML $\geq$4.7 | $th_{nDS}$, RML $\geq$4.7 |
| **EA-WR, 1h -** $GEV_3$ | **173** | 120 | **36** | 11 |
| **WeMOI, 1h -** $GEV_3$ | **248** | 150 | **50** | 17 |
| **EA-WR, 24h -** $GEV_3$ | **170** | 139 | **29** | 16 |
| **WeMOI, 24h -** $GEV_3$ | **213** | 168 | **43** | 23 |

**Table 4.** Number of doubly-stochastic sites (i.e., sites where RML$\geq th_{RML}$) in the original dataset ($n_{DS}$), and significance thresholds against spatial dependence (i.e., empirical and theoretical $th_{nDS}$). For each case, corresponding to a given duration and doubly-stochastic framework, the highest number between $n_{DS}$ and $th_{nDS}$ is marked with bold font.

Figure 6 illustrates for a specific location an example of doubly-stochastic modeling. Panels d-f show frequency curves corresponding to single "realizations" of the teleconnection of interest (i.e. WeMOI in the example, colorscale from yellow

to purple). Indeed, frequency curves are not theoretically consistent with the non-stationary framework. Nevertheless, the representation is very useful to visualize the degree of variability of design rainfall depth associated with alterations of the driving teleconnection.

As clearly illustrated in Figure 6, the variability of predicted rainfall percentiles associated with changes in the teleconnection may be very significant. It is worth noting that WeMOI changes in the example span across observed values, no extrapolation is performed. The prediction of the 100-year 24-hour rainfall depth in the selected location is equal to 240mm according to the stationary model ($GEV_0$), but may vary between 200 and 240 if the empirical relationship between WeMOI and L-CV is explicitly modelled ($GEV_2$), between 200 and 300 if the mean 24-hour annual maximum rainfall depth becomes a function of WeMOI ($GEV_1$), and may be as low as 170 and as high as 300 if both mean and L-CV are expressed as functions of WeMOI ($GEV_3$).

# 5    Discussion

## 5.1    Spatial correlation fields

The results in Section 4.1 clearly show that WeMOI and EA-WR have a very strong influence on the mean of extreme precipitation over the study area. This was expected, as it is in line with the observations of Caroletti et al. (2021) for Southern Italy, and Romano et al. (2022) for Central Italy. In fact, WeMOI consists of the normalized difference in atmospheric pressure between Cadiz, in the South of Spain, and Padua, in Northern Italy, and thus, it describes the formation of precipitation systems over the Tyrrhenian Sea (Lopez-Bustins et al., 2020; Redolat et al., 2019). Nevertheless, EA-WR pattern consists of four atmospheric anomalies extending from North Atlantic to Western Russia (Barnston and Livezey, 1987). Negative phases, which correspond to low-pressure anomalies over Europe and Northern China and high-pressure anomalies over the Central North Atlantic and North of the Caspian Sea, have been found to drive wetter conditions across Europe and the Mediterranean (e.g., Krichak and Alpert, 2005). Moreover, EA-WR is highly correlated with WeMOI, and hence is reasonable that the two teleconnections have similar influence.

In particular, the presence of significant intense negative correlation values in the Gulf of Genoa and the North-East for the mean with WeMOI (panels b and g of Figure 3) and EA-WR (panels d and i of Figure 3) is consistent with the known patterns of precipitation regimes over North-Central Italy. In fact, intense daily precipitation values are expected over the Tyrrhenian Coast and North-Eastern Alps in the presence of intense southwesterly flows from the Mediterranean, typical during the autumn season and favored by large scale circulation anomalies associated with negative value of WeMOI and EA-WR. In the remaining portion of the study area, precipitation systems are more complex, as influenced by the passage of cut-of lows favoring precipitation over the Southern and Eastern portion of the Apennines area. In this case, correlation patterns with WeMOI and EA-WR are expected to be more fragmented.

It is very interesting to notice that although the 1h rainfall maxima are mostly linked to convective phenomena, often characterized by a very limited spatial scale, the correlation with WeMOI and EA-WR presents more extended geographical patterns

than in the daily case (compare Figure 3.b and 3.d with 3.g 3.i).

Regarding the L-CV, the absence of evident correlation patterns for hourly annual maxima confirms our knowledge about spatial variability of convective phenomena (Figure 3.k-o). Small and fragmented, mainly positive patterns are visible for daily rainfall (Figure 3.p-t), but a physical interpretation of the L-CV-teleconnection dependence is more complex. In general, the consistency of patterns with spatial aggregation is lower than for the mean, which may be partly due to higher uncertainty in at-site computation of L-CV within a 10-year time window. For this reason, we believe that for L-CV, a spatial aggregation at 30km should be preferred for RFA.

An important element to consider when commenting Figure 3 is that only correlations with p-value $\leq 0.05$ were considered in the computation of $ri$ (eq. 7). Even though this threshold value is the most commonly adopted in the literature, it is still subjective, and it could lead to discard some significant correlations. This is evident when looking at the correlations with p-value within 0.1 and 0.05 in Figure 4 (i.e., tiles without red outlines). An additional source of subjectivity is the interpolation of $ri$ with kriging, that depends on the choice of kriging type and semivariance function.

Nevertheless, based on the results described in Section 4, the methods adopted generally lead to a useful characterization of the teleconnection-extreme rainfall correlation field, which can be used to drive frequency analyses. In fact, sub-regions with homogeneous rainfall-teleconnection dependence may be defined, in order to set specific doubly-stochastic RFA models. This option requires to define an objective grouping criterium, possibly based on a set of morphoclimatic characteristics (see e.g., the morphoclimatic characteristics adopted in Magnini et al., 2024); due its complexity, this problem deserves specific studies, and is not further investigated in the present study.

We estimate spatial correlation fields based on two major elements: temporal and spatial aggregation of the data. First, temporal aggregation through sliding time windows allows to consider the statistics of the extreme rainfall during time, instead of the rainfall depth themselves. In this way, it is possible to filter out inter-annual variability of the seasonality and magnitude of the annual maxima and focus on the decadal precipitation statistics. Second, spatial aggregation into tiles allows to obtain more reliable values of the rainfall statistics. This produces a smoothing local effect, that could be due to data fragmentation and noise and enhance geographical pattern recognition (see Figure 4). The choice of $w_t$ for spatial aggregation should be carefully conducted, as it mainly depends on the morphology of the study area, and the density and location of the rain gauges. Generally, considering a number of different values for $w_t$ is useful to analyze the reliability of the detected correlation patterns.

A different approach for spatial aggregation could be the one described in Castellarin et al. (2024), where overlapping tiles are used. However, including each gauged site in multiple tiles could result in eccessive smoothing of the orographic effect over the correlation field, and strenghtening of spatial dependence. Thus, this solution should be preferred only in case of scarce density of gauges network.

A key aspect of the proposed approach is its high adaptability. In fact, the same methodology with an appropriate parametrization could be used to study the influence of teleconnections on several environmental variables, such as AMS of floods or temperature or wind. Moreover, raster maps of the correlation field (as the ones in Figure 3) could be used as descriptors of the drivers of an environmental variable, and adopted as input of predictive raster-based models (e.g., for prediction of flood susceptibility, as in Magnini et al., 2023).

## 5.2 Doubly-stochastic regional frequency analysis

Looking at the results of the second phase of our study, two main points are of general interest. First, the range of variability of the expected percentiles with doubly-stochastic models (i.e., non-stationary models depending on teleconnections) is very wide, confirming what observed locally by other authors (Figure 6). Second, the regional dependency of rainfall statistics on teleconnections can be successfully exploited locally for frequency analysis (see Table 3 and Figure 5). This is a useful improvement over present literature, as it allows to obtain extreme rainfall statistics (i.e., $\mu_{st}$ and $L-CV_{st}$) even where observations are not locally available. Moreover, the observed variability of the expected maximum rainfall with given return periods could be used as teleconnection-informed uncertainty range for the design of hydraulic structures.

The framework we adopted for doubly-stochastic RFA is based on the strong assumption that the same type of function (i.e., a polynomial function) can represent the teleconnection-statistic relationships within all the study area. Indeed, this approximation is sufficiently accurate in some stations, while being not adatp in others. This is probably also the reason why $GEV_1$ models outperform $GEV_0$ globally more often than $GEV_2$ models (Table 3). In fact, the polynomial approximation may fit the data better when these are collected locally (as for the $\boldsymbol{\mu}_{\mathbf{tel}}$-$\boldsymbol{\mu}_{\mathbf{st}}$ case) than when they result from tile-wise averaging (as for the $\boldsymbol{\mu}_{\mathbf{tel}}$-$\boldsymbol{L}$-$\boldsymbol{CV}_{\mathbf{t}}$ case). Nevertheless, it is encouraging that a significant number of sites with RML$\geq$3 is detected for $GEV_1$ and $GEV_2$ (see Table 4), despite the low-complexity of the function adopted for modelling the dependence on teleconnections. Accordingly, $GEV_3$ models derive their goodness or badness from the sum of $GEV_1$ and $GEV_2$ contributes, which leads them to be the most adaptable models. Therefore, a high number of doubly-stochastic sites is observed when 1.6 is considered as RML significance threshold, corresponding to a small penalty for the additional parameters of $f(x)$ (compare 1.6 and 4.7 thresholds for $GEV_3$ in Table 3).

It is very interesting that the comparison of $ri$ (essentially based on Spearman correlation) and RML highlights differences in the most influent indices (i.e., compare Table 2 with Table 3) and doubly-stochastic patterns (e.g., compare panels k-t of Figures 3 and 5). A partial difference between the two metrics is expected and natural, mainly due to three reasons. First, $ri$ takes into account both the significance of the correlation and its sign, while RML represents the goodness-of-fit, and hence it does not describe the nature of the teleconnection-rainfall dependence (i.e., increasing/decreasing). Thus, RML may lead to group sites whose dependence on the teleconnection has different sign. Second, RML is probably more similar to a single correlation (i.e., at-site in the $GEV_1$ case, and at 30km-tiles in the $GEV_2$ case), while $ri$ depends on correlations at several spatial aggregations, which may disagree. Third, also correlations that are not considered significant could lead to good doubly-stochastic models. Furthermore, some studies (e.g., Sugihara et al., 2012; Cappelli and Grimaldi, 2023) suggest that correlation may fail in describing the causality of two variables, and that an approach based on a model's performance can be more accurate. However, we believe that in our case both the metrics are very useful, as they can help to delineate sub-regions with homogeneous teleconnection-rainfall dependence function (see also Section 5.1). As perfectly exemplified in our results, sometimes the presence of such regions is evident (e.g., the Gulf of Genoa and North-East for $GEV_1$, see Figure 5), while in other cases it is hard to distinguish between small homogeneous regions and effects of spatial dependence (e.g., Adriatic coast and western Gulf of Genoa for $GEV_2$, see Figure 5).

It is important to underline that the aim of the present research is to investigate the potential of teleconnections as independent variables in RFA models, and not to propose a specific method for RFA. Indeed, the RFA results depend on a number of parameters, including the widths of sliding time windows for temporal aggregation of the teleconnection indices ($w_{tel}$) and AMS ($w_{AMS}$), and the resolution for spatial aggregation ($w_t$). These can be set after a careful sensitivity analysis for defining the spatial field of the teleconnection-statistic correlation (see Sections 2 and 4.1). The case of formalizing a function $f(x)$ of extreme rainfall statistics depending on teleconnections is very different and much more complex. In fact, one should decide not only the shape of this function, but also the way its parameters vary in space and should be estimated, which may require the delineation of sub-regions (see considerations above). In our study, we adopted a simple framework, as this function has a limited number of parameters and the same shape (i.e., polynomial) in all the spatial domain. We showed a hierarchical RFA approach where the parameters of the polynomial functions are fitted at-site for $\mu$ and at 30km-tiles for L-CV. Our analyses overall suggest that even with a simple RFA framework, the use of teleconnections as dependent variables to describe the extreme rainfall regime may increase the accuracy in frequency modelling.

Different approaches are indeed possible. First, the best resolution for spatial aggregation and the shape of the teleconnection-statistic function should be carefully evaluated for each specific case. Second, a more complex teleconnection-statistics function could be defined. A possible approach is the one proposed by Magnini et al. (2024), which leverages neural networks' capabilities to obtain functions whose parameters depend on the location of the considered site and other morphoclimatic descriptors, or the one adopted by Machado et al. (2015), which exploits generalized additive models. Indeed, the implementation and discussion of more sophisticated RFA methods to exploit teleconnections' informative content is complex, and should be addressed by future studies.

## 6 Conclusions

A growing number of recent studies show how large scale climatic indices (or teleconnections) can be used as covariates to increase reliability of local frequency analysis of rainfall extremes across diverse geographical regions worldwide (e.g., Fauer and Rust, 2023; Ouarda et al., 2020; Ragno et al., 2018). It is theoretically possible to extend these methods to regional frequency analysis (RFA), but the teleconnection-extreme rainfall dependency at a regional scale should be first investigated. Beside its usefulness for correct estimation of the design rainfall for engineering applications, this topic is still not well addressed in the literature.

In the present study, we propose a framework to assess the link between teleconnections and the frequency regime of rainfall extremes at a regional scale, in order to perform climate-informed RFA. The approach is tested for a large and climatically diverse region in Northern Italy. Our dataset consists of 680 annual maximum series (AMS) of hourly and daily (i.e., 1 and 24 hours durations) rainfall depth, recorded between 1921 and 2022. We select six climate indices, known to have significant correlation with local climate variability over the study area (Caroletti et al., 2021; Criado-Aldeanueva and Soto-Navarro, 2020; Romano et al., 2022): the North Atlantic Oscillation, Pacific Decadal Oscillation (PDO), East Atlantic – West Russia pattern,

El Niño Southern Oscillation, Mediterranean Oscillation Index, and Western Mediterranean Oscillation Index (WeMOI).

The main steps of the proposed framework can be summarized as follows. First, we define sliding time windows in order to obtain time series of pluridecadal averaged teleconnections and statistics of annual maxima. The latter consist of the sliding mean and L-coefficient of variation (L-CV) of AMS, which in our case are used to characterize the distribution of sub-daily rainfall extremes. Second, we discretize the study area into tiles where L-moments are averaged into regional predictions. Then, we evaluate the correlation of teleconnections with time series of spatially gridded L-moments. Finally, we show a preliminary application of climate-informed RFA of rainfall extremes, where L-moments are modelled as functions of the teleconnections singularly. These models are compared with the corresponding stationary models by means of goodness-of-fit metrics.

Our results show that the sliding mean of annual maxima (both hourly and daily) has a higher number of significant correlations with WeMOI and EA-WR than with the other indices. Moreover, the relationship between these indices and sliding mean of extreme rainfall shows clear spatial patterns across the study area, whose robustness is confirmed by their limited sensitivity to the chosen grid resolution and the partial agreement with previous studies (Caroletti et al., 2021; Romano et al., 2022). Similar patterns are found when considering the areas where the goodness-of-fit of the climate-informed regional models (where the mean depends on the WeMOI and EA-WR) outperforms the stationary approach. As well, this is coherent with the known spatial variability of precipitation regimes over the region.

The relationship between the sliding L-CV and the teleconnections is more complex, since the number of significant correlations is lower, and dependence patterns are less extended and more heterogeneous. Nevertheless, the regional models where L-CV depends on the teleconnections outperforms the stationary approach in a significant number of stations.

The proposed approach is simple and easily reproducible, yet it is new with respect to the existing literature. In fact, while most authors investigated the correlation between the teleconnections and the raw AMS, we consider the L-moments. This, in combination with spatial discretization of the domain, allows us to focus on the relationship between the teleconnections and the extreme rainfall regime, instead of the extreme values themselves, whose seasonality and interannual variability can affect the correlation analysis. Beside the preliminary nature of our RFA application, commonly used metrics (e.g., Ashkar and Ba, 2017) detect overall an increase in goodness-of-fit with respect to a stationary approach, in line with previous studies (Nerantzaki and Papalexiou, 2022). This shows that teleconnections may be useful covariates in a regional a framework.

Overall, our research suggests promising pathways for climate-informed local and regional frequency analysis of rainfall extremes, and our methodology is highly adaptable to different environmental variables, such as floods and temperature.

*Data availability.* The rainfall data used in the present study are part of the dataset I2-RED (Mazzoglio et al., 2020), available under authorization at the website https://doi.org/10.5281/zenodo.4269509. Teleconnections indices data are available at https://psl.noaa.gov/data/climateindices/list/ and https://crudata.uea.ac.uk/cru/data/.

## Appendix A:  Statistical significance of empirical RML values: Monte Carlo simulations

The present Appendix describes the methodology adopted to empirically assess the statistical significance of RML values obtained in the study (i.e., ratio of models' likelihood, see Section 2.2). In particular, the statistical test consists of comparing RML with lower threshold values ($th_{RML}$) associated with a 5% significance level (see Section 2.2). We determine these threshold values by means of Monte Carlo (MC) simulations that are designed to empirically identify the RML value associated with a p-value of 95% when comparing the likelihood of a doubly stochastic GEV (i.e., $GEV_{DS}$ with DS=1, 2, 3, see also Section 2.2) with that of a stationary distribution (i.e., $GEV_0$) when the parent distribution of the generated annual sequences is $GEV_0$. Specifically, our simulations generate stationary synthetic annual sequences (AMS), but we do not fit new frequency distributions on the synthetic AMS. Instead, the original frequency models are used to compute the likelihood of the synthetic timeseries. The procedure to define values of $th_{RML}$ can be described as follows:

1. For each station, the $GEV_0$ model (i.e., the stationary distribution fitted to the at-site mean, and regional L-CV and L-CS) is used to generate 1000 annual maxima time series with the same length of the original AMS.

2. For each simulation, RML is computed with reference to the $GEV_0$ model and, in turn, one of the considered doubly-stochastic models $GEV_{DS}$ (i.e., $GEV_1$, $GEV_2$, and $GEV_3$). As stated above, we consider only the frequency models fitted to the original data, but RML refers to the synthetic time series. Since the parent distribution is $GEV_0$, RML is expected to be low (i.e., $GEV_0$ should provide the best fit in the majority of cases).

3. For each station ($st$) in the study region and each doubly stochastic GEV model, we identify the 95th percentile from the series of simulated RML. This step produces 680 percentiles for each doubly stochastic model, meaning that each station has its own percentiles ($th_{RML,st}$).

4. The values of $th_{RML}$ for each doubly-stochastic GEV model for the entire region are defined as 95th percentiles of the 680 at-site values $th_{RML,st}$ from the previous step. Thus, we obtain a single RML threshold for a specific $GEV_{DS}$ model that is valid for the whole study region.

We repeat this procedure for both durations, 1h and 24h, and all of the teleconnections considered for the test, namely NAO, EA-WR, PDO, WeMOI and MOI (see Section 4.1). We therefore obtain several values of $th_{RML}$, each one corresponding to a specific duration, teleconnection and GEV type.

The procedure described above preserves, or reproduces, all the main elements of our specific application case that may impact RML's diagnostic power: the study area morphoclimatic variability is considered by applying the MC experiments to the entire dataset; synthetic time series preserve the original length at each site (step 1). Furthermore, considering the original frequency distributions (step 2) is a key point, as it allows us to test the predictive power of RML in the real case, avoiding us to make additional assumptions on synthetic distributions.

Having the MC reference threshold values of RML, we can test the doubly-stochastic hypothesis; if the RML value we obtain for a specific case in our study region (i.e., for a given site duration, teleconnection and GEV model) is greater than the corre-

490 sponding simulated $th_{RML}$, then the frequency regime of rainfall extremes for that site can be considered doubly-stochastic at 5% significance level.

**Appendix B: Field significance of doubly-stochasticity signals: spatial bootstrap resampling**

This Appendix describes the bootstrap experiments adopted to test the field significance of doubly-stochasticicty spatial signals in presence of cross-correlation, or intersite spatial dependence (see Sections 2.2 and 3.2 and 4.2). In particular, we test the 495 statistical significance of the number of doubly stochastic regimes that is observed in a specific region ($n_{DS}$), that is the number of sites where, according to the RML analysis, a given $GEV_{DS}$ for a specific teleconnection provides a better fit to the data relative to $GEV_0$. The test adopts a threshold number of stations, $th_{nDS}$, that represents the 95th percentile of 1000 $n_{DS}$ realizations for 1000 stationary (i.e. non doubly-stochastic) synthetic regions, which are generated from the original regional sample of annual maxima through spatial bootstrap resampling (see also Castellarin et al., 2024; Vogel et al., 2001). The 500 bootstrap procedure to define values of $th_{nDS}$ is the following:

1. We generate 1000 synthetic regions by randomly reshuffling year-wise 1000 times the dataset of 680 AMS of rainfall depths with duration 1h (or 24h) year-wise. Each reshuffling produces a synthetic AMS dataset that is distinct from the original one, while preserving the distribution of observations among all AMS in any given year. Since the reshuffling is random, the 1000 synthetic datasets are realizations of a stationary world.

2. For each synthethic reshuffled dataset, we repeat the same hierarchical regional frequency analysis (RFA) adopted for the original dataset (see Sections 2.2 and 3.2). First, the stationary model, $GEV_0$, is fitted to the syntehtic data. Second, the synthetic time series of at-site $\mu$ and 30km tile L-CV are computed. Third, the polynomial functions are fitted to the synthetic $\mu$ and L-CV timeseries and the original (i.e. non-reshuffled) teleconnection timeseries; it is worth underlining here that any correlation, or relationship, between the original teleconnection and the synthetic $\mu$ and L-CV timeseries 510 is spurious, as, by construction, these synthetic $\mu$ and L-CV timeseries are obtained from stationary realizations. Fourth, the parameters of $GEV_1$, $GEV_2$ and $GEV_3$ are obtained for each synthetic time series.

3. For each synthethic reshuffled dataset, RML values (ratio of models' likelihood, see Section 2.2) are computed for $GEV_1$, $GEV_2$ and $GEV_3$

4. For each synthethic reshuffled dataset, $n_{DS}$ is computed for $GEV_1$, $GEV_2$ and $GEV_3$ (see also Appendix A and Section 515 3.2)

5. The values of $th_{nDS}$ are then obtained as the 95th percentiles of 1000 synthetic $n_{DS}$ obtained for each considered case, that is given duration (i.e., 1h or 24h), teleconnection (i.e., EA-WR and WeMOI, see Section 4.2), doubly stochastic model $GEV_{DS}$ (i.e., $GEV_1$, $GEV_2$ or $GEV_3$), and RML threshold (i.e., the theoretical one, and the empirical obtained using the procedure illustrated in Appendix A).

The field significance of the doubly-stochasticity signals can then be assessed at the 5% significance level by testing whether the empirical $n_{DS}$ values obtained for the original dataset are larger than the empirical $th_{n_{DS}}$ obtained as described above.

*Author contributions.*  AM: conceptualization, investigation, methodology, software, writing – original draft, writing – review and editing. VP: methodology, writing – original draft, writing – review and editing, validation. AC: conceptualization, methodology, writing – original draft, writing – review and editing, validation, supervision.

*Competing interests.*  The author declare they have no known competing financial interest or personal relationship that could have appeared to influence the work reported in this paper

*Acknowledgements.*  The work was supported by European Climate, Infrastructure and Environment Executive Agency (CINEA), under grant number 101069928 — LIFE21-IPC-IT-LIFE CLIMAX PO. The Authors thankfully acknowledge the use of free and open-source software, in particular Python (Van Rossum and Drake, 2003) and R (R Core Team, 2024). Also, sincere gratitude is due to Paola Mazzoglio and 530   Pierluigi Claps, for giving access to the dataset I2-RED (Mazzoglio et al., 2020).

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

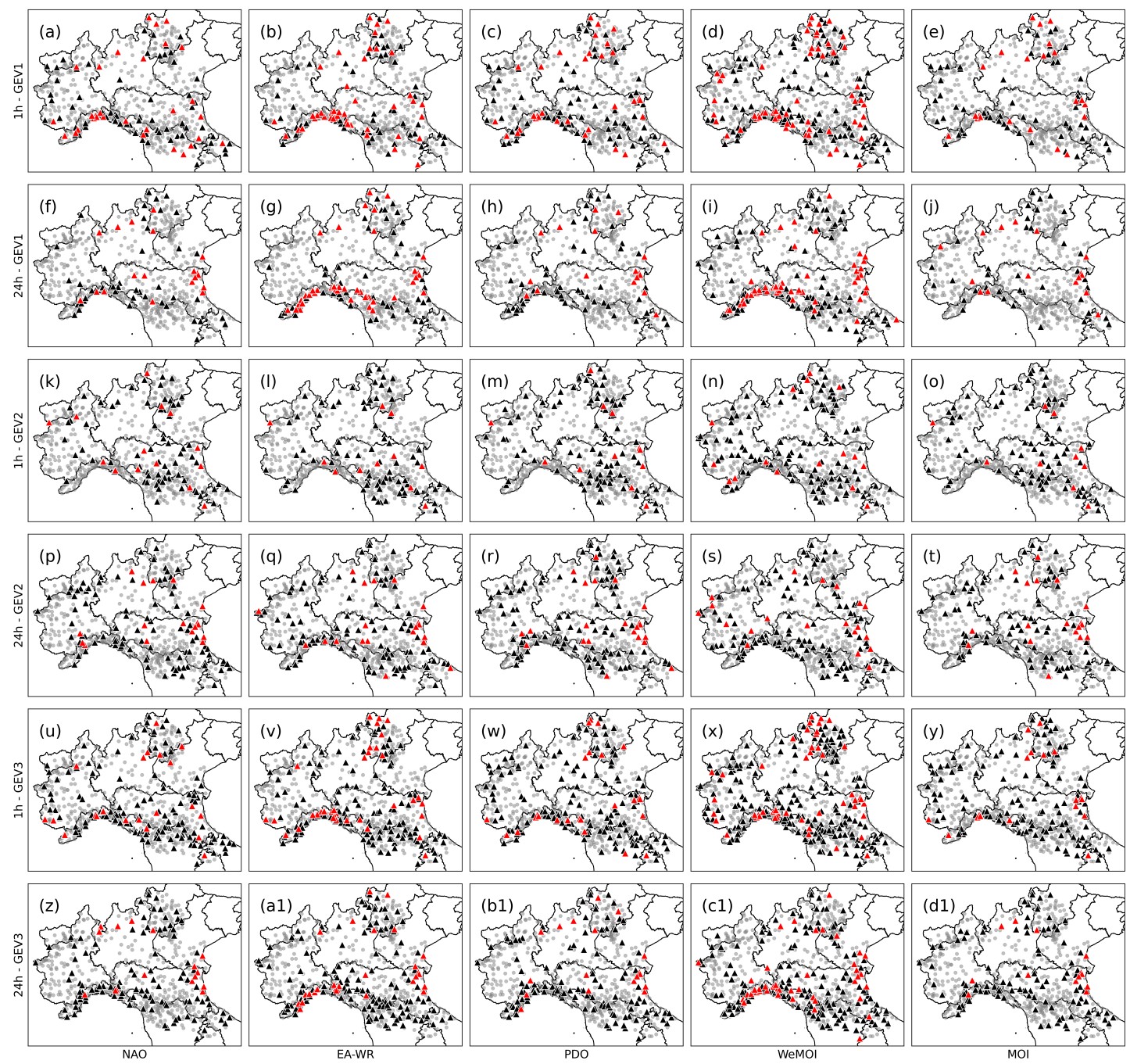

**Figure 5.** Doubly-stochastic sites observed for $GEV_1$ (panels a-i), $GEV_2$ (panels l-o) and $GEV_3$ (panels u-(d1)). Black triangles represent sites where theoretical $th_{RML} >$RML$\geq$empirical $th_{RML}$. In red triangles, RML$\geq$ theoretical $th_{RML}$. Stations where RML$<$ empirical $th_{RML}$ (i.e., stationary sites) are represented with grey dots.

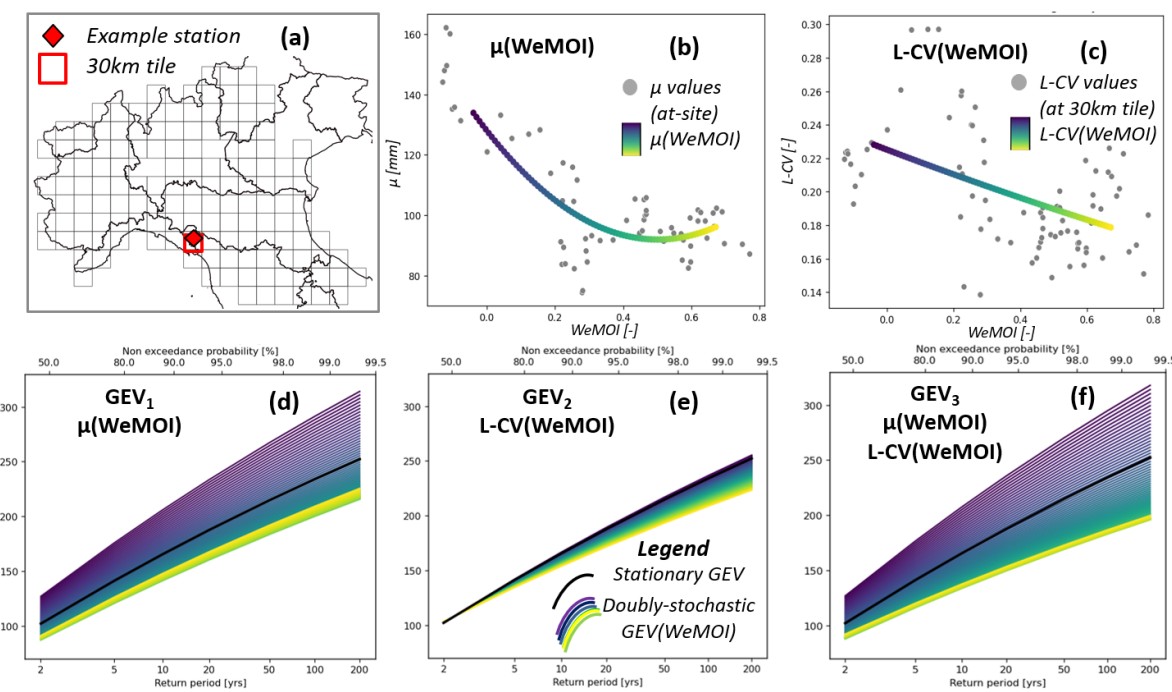

**Figure 6.** hierarchical frequency analyses of 24h rainfall annual maximum series at a given location where the hypothesis of doubly-stochasticity in the mean ($\mu$) and L-CV resulted to be significal at 5% level. Upper panels: considered station (a), polynomial function of $\mu$ (b) and L-CV depending on WeMOI (c). Lower panels: expected percentiles with given return periods in stationary (black line) and doubly-stochastic framework (colored scale lines), under the assumption that the mean depends on WeMOI ($GEV_1$, panel d), L-CV depends on the WeMOI ($GEV_2$, panel e), or both mean and L-CV depend on the WeMOI ($GEV_3$, panel f).