# Peer review of "Informativeness of teleconnections in frequency analysis of rainfall extremes"

_EGUsphere, 2024_

## Author Response (AR1)

**Rebuttal letter - Manuscript 2024-3261**

Dear Associate Editor,

The attention and consideration you dedicated to the revision of our manuscript, as well as your gentle assistance, are extremely appreciated.

The comments and suggestions we received are very significant, and guided us through a critical revision of our methods, results, and discussion. Accordingly, these three parts have been strongly modified in terms of figures and text in the second version of our manuscript. Furthermore, two Appendixes have been added, to describe additional analyses we performed. However, we would like to underline here that the aims and approach of our study, as well as our main conclusions have not changed.

We report our replies to the three reviewers below.

**Francesco Serinaldi**

**F. S.:** *I would like to share some thoughts about this paper with the Authors, hoping that they can contribute to the discussion.*
*A fundamental assumption that is common to almost all methods for regional frequency analysis is spatio-temporal independence. However, the proposed procedure seems to neglect it and introduces a spurious dependence as well, I think. In fact, the sliding window procedure used to compute the sequences of L-moments acts as a moving average (… it is the same procedure used to compute e.g. drought indices such as SPI or SPEI). The resulting time series have a spurious autocorrelation with linear decay of 1/w per time lag. It is known that the autocorrelation affects the estimation of cross-correlation (and vice versa). It can yield spurious cross-correlation, and variance inflation. This means that the sequences of L-moment values computed over sliding (overlapping) time windows and the "rolling mean of the considered teleconnection" might show a spurious cross-correlation. Also note that WeMOI is a low frequency climatic mode characterized by its own "natural" autocorrelation. Therefore, the autocorrelation of the rolling means of WeMOI are characterized by the superposition of two autocorrelation structures.*

*In this context, any statistical test used to check the statistical significance of cross-correlation values should account for the variance inflation affecting the distribution of the test statistics (here, the Spearman correlation; see e.g. works by Khaled H. Hamed in this respect). I may have missed something, but the text seems not to specify whether the*

*Authors accounted for these issues. If not, I think they should be considered, as they often completely change the conclusions of these types of analysis, revealing that dependence might be a much more general and simple way to model the observed behaviour (... and "entia non sunt multiplicanda praeter necessitatem").*

> **Reply of the Authors:** Thank you for this important and relevant comment. The potential impact of spurious autocorrelation was not considered in the analysis. We have repeated our analyses by adopting the Spearman correlation coefficient with the appropriate significance test, as provided by the R library *corrTESTsrd* (Lun et al., 2023; see Section 2.1). Our results (Section 4.1) show that the number of significant correlations detected for WeMOI is no longer overly higher relative to other indexes. However, these results overall agree with our previous analyses, suggesting that (1) several significant correlations are present, and (2) some spatial correlation patterns between extreme rainfall statistics and climate indexes exist (see Sections 4.1, and 5.1).

*__F. S.:__ The Authors denote the GEV models with parameters depending on WeMOI as nonstationary. Nowadays, the term "nonstationary" is used quite arbitrarily and loosely in almost every paper; however, a model is nonstationary if and only if its form (parameters) depends on a parametric support such as time or space. WeMOI is not a parametric support; it is a process with stochastic behaviour (and periodic oscillations at 12 months, and about 20 and 50 years). Models with parameters depending on other "random" processes are not nonstationary but doubly stochastic because the parameters are themselves randomly fluctuating.*

*Please note that this is not just a semantic issue. Double stochastic models can be stationary, thus meaning that we can apply all standard results of mathematical statistics. Conversely, nonstationary models might be problematic and lead to paradoxes and misleading conclusions as they are not consistent with the ergodicity assumption, which is fundamental to establish a correspondence between sample properties and population properties, thus making inference technically possible.*

> **Reply of the Authors:** Thank you for this comment. Our original manuscript uses the term "non-stationary" in a way that is similar to many publications (see examples and literature in "The legacy of STAHY" by Volpi et al., 2024). We agree with you on the fact that "doubly stochastic" might be a better characterization of the models we are testing in our analyses (as also suggested in Serinaldi and Kilsby, 2018), andhte revised version of our manuscript refers to the suggested terminology (e.g., see Section 2.2).

**F. S.:** *An example of the consequences of neglecting the importance of underlying assumptions is the interpretation of the performance metrics used to compare stationary and nonstationary models. The RLM index in eq. 6 is related to the test statistic of the likelihood ratio test*

*$Chi^2 = -2*ln(LH\_0/LH\_1) = -2*ln(LH\_st/LH\_nst) = 2*ln(LH\_nst/LH\_st) = 2*RLM$,*

*which is asymptotically distributed as a Chi^2 variable with degrees of freedom equal to the difference of the number of parameters of the two models. Since the Authors use second-order polynomials for mu and/or CV, the degrees of freedom are 2 (=5-3) for GEV_1/2 vs GEV_0 or 4(=7-3) for GEV_3 vs GEV_0. Therefore, the 95th quantiles of the two Chi^2 distributions are about 6 and 9.5, corresponding to critical levels of RLM equal to 3 and 4.25. If we compare these values with those reported in Fig. 6a, we see that most of the RML values are lower than those upper limits. Leaving aside the validity of the asymptotic Chi^2 for finite samples, the message is that relative measures/metrics should not be interpreted according to their absolute values but should be assessed accounting for their own sampling variability. Indeed, this interpretation reconciles results of Fig. 6a with those in Fig. 6c.*

> **Reply of the Authors:** We totally agree with the fact that the test statistics should not be considered alone, but with their associated significance value. However, we are not totally persuaded that the reference value of the RML metric you suggested should be used in our analysis. This is mainly due to two reasons: (1) the size of our samples is limited and (2) the nature of the parameters of the polynomial function is different from the other GEV parameters. To share some light on this aspect, we performed Monte Carlo resampling experiments that are specifically designed for assessing empirical significance values for RML. Our results seem to indicate that a lower reference value should be used for identifying statistically significant improvements over a stationary model (see Sections 2.2 and 3.2 and Appendix 1 of the revised manuscript).

**F. S.:** *In this respect, please note that TN.SW is the only theoretically consistent goodness-of-fit metrics used here, whereas AD is not.*

*Let me explain. When we deal with (supposed) nonstationary models we can only apply diagnostics to standardized versions of the data, conditional on the fitted parameters (see e.g., Coles 2001; An Introduction to Statistical Modeling of Extreme Values). The expression of AD statistic holds true if and only if the data are identically distributed (i.e. the model is stationary) because such a goodness-of-fit test is based on the distance between the*

*parametric model and the empirical cumulative distribution function (ECDF). Now, the values of the ECDF, F_n(x), corresponding to each observation are representative of theoretical probabilities if and only if these observations come from the same distribution F(x), thus meaning that larger observations (order statistics) correspond to larger non-exceedance probabilities.*

*However, suppose that our model is nonstationary; for example, GEV location parameter increases with time because the observations seem to assume greater values through the years. In this case, each observation comes from a different distribution (which is only valid at a specific time... or WeMOI value), and largest values are likely associated to GEV distributions with large location parameters (as the time-varying GEV model attempts to follow the fluctuations of the observations). Therefore, given a sample of size n, the largest observation, for instance, has non-exceedance probability 1/(n+1) according to the ECDF (... leaving aside plotting position issues), whereas it might have the same probability of a smaller observation under its own local GEV. In other words, the correspondence between empirical and theoretical probabilities has been lost. Therefore, Delta_AD in Fig. 6a is positive for two reasons:*

- *AD_nst is likely greater than AD_st because the ECDF appearing in the AD formulas refer to (overlooked) stationary assumption.*
- *The AD statistic is a positive distance that indicates better performance when it is small (but not too small, is it would mean that the F(x_i) sample is too regular to be a random uniform sample); if Delta_AD = AD_nst – AD_st is positive, it means that AD_nst is larger than AD_st, and therefore GEV_0 is better. In fact, this is the rule used by Ashkar and Ba (2017): "The decision rule for the sample is to choose GP if a_GP– a_KAP < 0".*

*That said, as for RLM, the actual question is whether values of Delta_AD equal to 0.004-0.006 are just within the expected fluctuations for nested models, once we account for factors like dependence and the variance inflation of AD statistics related to the fact that the model parameters are estimated on the same data used to compute the AD statistic itself. Since estimated parameters make AD statistic no longer distribution-free, comparing AD values of different models is also questionable (because identical AD values for two models can correspond to different probabilities in the sampling distributions of AD under these alternative models).*

> **Reply of the Authors:** Many thanks for this useful and relevant comment. Based on these considerations, we decided not to adopt the AD metrics in the revised version of our study (see Section 2.2 of the revised manuscript).

*F. S.: TN.SW values confirm the message of the other two metrics: the data are not enough to discriminate among GEV_0, GEV_1, etc. However, while RLM and AD and their Delta's are expected to be positive but possibly small, SW is centered around zero by construction (if a model is good enough). Therefore, Delta_TN.SW is expected to be centered around zero when discriminating between models is not possible. In this case, the comparison is technically sound because TN.SW provides the kind of conditional standardization mentioned above. Thus, results in Fig. 6c are consistent with those in Figs. 6a and 6b, and the interpretation of Fig. 6 provided in section 5.2 should be revised accordingly, I think.*

> **Reply of the Authors:** We understand the reasons why the TN.SW metric may be considered theoretically consistent with the framework adopted in our study. However, additional analyses inspired by your comments and conducted with the same methodology as for RML (see Appendix A in the revised manuscript), led us to question its true validity and informativeness for our specific study. More specifically, a series of Monte Carlo experiments seem to suggest that TN.SW has a low discrimination power in our case. Thus, TN.SW is not adopted in the revised study.

*F. S.: A note about the use of return period: return period is the expected value of return intervals, which implies integration over time (by definition). Under nonstationarity, the integral does not yield 1/(1-F(x)) because a unique F(x) does not exist! And replacing it with a set of T_i = 1/(1-F_i(x)) formulas makes no sense. Roughly speaking, under nonstationarity, integrating over time implies averaging over a set of F_i(x) distributions, and the resulting expectation is a formula reflecting (and function of) all F_i's. While I understand the (fallacious but widespread) rationale of drawing a set of return level curves (as those in Fig. 5c-e), the formula 1/(1-F_i(x)) does not correspond to any expected value (over time). Under nonstationarity, the return level curve is as unique as in the case of stationarity because the expected value of the (inter)arrival times of exceedances over a specified threshold is always a single value resulting from integration. However, what changes is the expression linking T with the set of F_i's. Even though the diagrams of T_i = 1/(1-F_i(x)) vs x are reported in almost every paper dealing with nonstationary distributions, this does not make them and the relationships T_i = 1/(1-F_i(x)) meaningful. So, please consider avoiding further spreading such a misconception.*

> **Reply of the Authors:** Thank you for raising this point. Indeed, we agree with you about the theoretical nonexistence and incorrectness of a single $T_i = 1/(1-F_i(x))$, and yet we disagree with you on the argued meaninglessness of the set of curves. Quite the opposite, we believe that this figure is very informative as it clearly shows the variability in terms frequency regime of extreme rainfall events that is associated

with different climatic conditions, described in our case by the value of WeMOI averaged in the last 30 years. This is very useful from a practical viewpoint in engineering practice for defining possible meaningful future climate scenarios. Nevertheless, we duly noted the theoretical limitations of such a representation, which are clearly mentioned in the revised version of our manuscript (see Section 4.3).

*F. S.: Finally, the Authors state that "the spatial aggregation into tiles allows to obtain more reliable values of the rainfall statistics". This is strictly true if the time series within a tile are independent; otherwise, spatial dependence implies information redundancy, meaning that the apparent smoothness comes with uncertainty much larger than one can think, and such averaged/aggregated statistics might be not so reliable. Please note that I do not refer to the spatial dependence of AMAX: these can look approximately uncorrelated (in space and time) even when the underlying processes are strongly dependent. In these cases, clustering in space and time might be an indicator of the underlying (concealed) spatio-temporal dependence. These remarks apply to any type of analysis, including for instance record-breaking observations. In fact, "significant deviations in the number of record-breaking events in an observed time series relative to what is expected under the iid hypothesis indicate non-stationary time series" (Castellarin et al. 2024; https://doi.org/10.3390/atmos15070865)... or dependence, I would say! If "I.I.D." still means independent and identically distributed, discrepancies can come from lack of fulfilment of one of these two assumptions or both, and there is no reason to exclude the former. Based on my experience, people often tend to neglect dependence because adding a few covariates to GLM-like models with an arbitrary polynomial/spline link is a bit easier than deriving the theoretical relationships accounting for dependence.*

*Overall, in my opinion, any method that implies spatio-temporal aggregation, smoothing, and averaging of hydro-climatic data should carefully account for spatio-temporal dependence, as this assumption allows one to keep models simple and parsimonious, it is generally sufficient to describe the behaviour of most of the observed processes, and often reveals that we are overconfident about the reliability of results and the amount of information (effective sample size) actually available.*

> **Reply of the Authors:** We are aware that spatial correlation among the time series of annual maxima increases the uncertainty of regional estimations. However, it does not introduce bias (see e.g., Hosking and Wallis, 1988), and thus we believe this aspect has a very limited impact on our analyses, since we adopt spatial discretization only to obtain estimations of the higher order L-moments (and consequently, of the quantiles) that are locally more robust. On the fact that

regional frequency analysis should be preferred to local frequency analysis, the classical literature is clear (see refs. above). Nevertheless, we agree with you that intersite cross-correlation could inflate the number of doubly-stochastic signals in our study area. Thus, we define and perform an ad hoc significance test (see Sections 2.2, 4.2 and Appendix B).

**F. S.:** *However, if we decide to use nonstationary models, we must bear in mind (i) what nonstationarity technically means and implies, (ii) that most of the methods available under stationarity are no longer valid, and (iii) the inference procedure itself along with the interpretation of results are problematic because of lack of ergodicity. The foregoing discussion just mentions some concepts that cannot be transferred when we move from stationarity to nonstationarity. Deeper discussion of these and other issues can easily be found in the literature... some of such a literature (concerning the impact of spatio-temporal dependence on frequency analysis) is from one of the Authors.*

> **Reply of the Authors:** We are particularly glad that the topics discussed in our manuscript inspired such a rich, useful and interesting comment. However, we believe that some of these points are not particularly relevant to our analyses. We agree that the utilization of nonstationary (or doubly stochastic) approaches implies the redefinition (or the careful selection) of some methods often adopted with stationary analysis (e.g., the goodness-of-fit metrics, as suggested here or trend tests, as shown in Serinaldi, 2024). However, recent literature urges for approaches and methods that can better capture the effects of climate variability on the frequency of hydrological extremes (e.g., Volpi et al., 2024, and all the references contained there). Hence, the identification of reliable and informative nonstationary frequency models does not seem to be a matter of choice anymore, but rather an open research avenue.

> That said, our study does not aim at proposing specific nonstationary models, but rather at investigating the regional characteristics of the link between teleconnection and extreme rainfall regime. What is the best way to implement doubly stochastic RFA models and how to correctly use them for statistical inference are definitely key topics, but they need to be addressed by future studies (see Section 5.2).

**Anonymous Referee #1**

**Referee #1:** *The article "Informativeness of teleconnections in local and regional frequency analysis of rainfall extremes" proposes a framework to assess the informative*

*content of teleconnections for representing and modeling the frequency regime of rainfall extremes at a regional scale, using northern Italy as a case study.*

*The article deals with a topic still poorly investigated, i.e. the possible use of information related to teleconnection in regional frequency analyses. Overall, the article is well written, the methodology is clearly described, the parameter selection has been properly described and the results are supported with evaluation metrics and a complete and detailed discussion.*

*Considering its novelty, I believe the article can be published after minor revision. It would be interesting to read what I suppose will be the next article, i.e. the application of the regional frequency analysis and a comparison with other methodologies.*

> **Reply of the Authors #1:** We are extremely glad that you appreciated our methodology and discussion. Moreover, we believe that the additional analyses inspired by the other comments received will further improve the relevance and robustness of our investigations.

***Referee #1:*** *A list of minor comments follows.*

*In the description of Figure 2, I suggest replacing "In black: Italian administrative regions" with something similar to "Black lines: Italian administrative regions" to avoid confusion with the color scale of the elevation.*

> **Reply of the Authors:** Many thanks for this useful point. The revised version of our article follows this suggestion.

***Referee #1:*** *Row 157: I suggest replacing "4000 m a.s.l." with "about 4000 m a.s.l.".*

> **Reply of the Authors:** We agree that the suggested wording is more accurate. The revised version of our article follows this suggestion.

***Referee #1:*** *Row 163: I suppose an "of" is missing after "time series".*

> **Reply of the Authors #1:** Indeed, this is a typo. Many thanks for highlighting it.

***Referee:*** *Row 223: Please check the reference, I suppose it should be "Gabriele and Arnell (1991)".*

> **Reply of the Authors #1:** Many thanks for raising this point.

***Referee #1:*** *In Figure 3, first column, some stations are colored with gray, while others are in a purple-green-yellow color scale. Could you please explain why? I suppose that you reported with this color scale only the 387 stations with a significant correlation with*

*WeMOI, but maybe I am missing something. Does the same apply to the white tiles in columns 2-4 of Figure 3?*

> **Reply of the Authors:** We are very grateful that you pointed out that this description is missing. Your interpretation is correct, and this aspect is clarified in the revised manuscript.

*Referee #1: In row 165, the authors said that the data covers the period 1928-2020, while in the conclusion (row 362) the period mentioned is 1921-2022. Please check and correct.*

> **Reply of the Authors:** Many thanks for highlighting this inconsistency. This will be corrected in the revised manuscript.

**Anonymous Referee #2**

*Referee #2: The manuscript presents a methodological framework for investigating the relationship between teleconnections and the frequency regime of extreme rainfall at a regional scale, applying it to a climatically diverse area in northern Italy. By analyzing six global climate indices, the approach assesses their correlation with the L-moments of annual maximum rainfall series at hourly and daily scales. The findings indicate that the WeMOI index exhibits the strongest connection to extreme rainfall, showing clear spatial patterns consistent with the region's precipitation variability.*

*The proposed topic is particularly relevant, and the implemented methodology is innovative and of potential interest to a large part of the scientific community. The manuscript is well-written, accurate, and includes an extensive case study that supports the general conclusions.*

> **Reply of the Authors #2:** We are extremely glad that you appreciated our methodology and innovative aspects. Moreover, we believe that the additional analyses inspired by the other comments received have further improved the relevance and robustness of our investigations.

*Referee #2: I am in favor of its publication, with only one suggestion regarding the abstract: it should be rewritten to be more accessible and communicative, allowing a broader audience to better understand the aim, methodology, and results.*

> **Reply of the Authors:** Many thanks for your positive opinion and the suggestion. The abstract has been extended and enriched.

Overall, we totally agree with the Reviewers that some important points needed to be discussed, and some elements needed to be improved or corrected. As mentioned above, we have been intensively working to refine and complete our analyses. More specifically, these additional investigations aim at (1) addressing autocorrelation issues when computing teleconnections-rainfall correlation, (2) understanding the statistical significance of our metrics, and (3) assessing the significance of doubly-stochasticity signals in presence of intersite dependence. We would like to underline that our main results seem to confirm most of our previous findings. Thus, we believe that our research questions have universal interest, our methodology is valid (and now improved thanks to the useful comments received), and our results are coherent with our current knowledge about the frequency regime of extreme precipitation over the study area.

Furthermore, we modified other elements of the manuscript in order to make it more readable and comprehensible. Apart from slight rephrasing throughout the text, these modifications include the reorganization of the "Parameterization" part, which is now a subsection of Section 3 (i.e., 3.2), instead of a subsection of Section 4 (i.e., 4.1). Also, in the parameterization Section some formulas have been omitted for the sake of brevity and clarity. Finally, the title has been slightly modified, as additional discussion about our methods and results lead us to conclude that the new version (i.e., "Informativeness of teleconnections in frequency analysis of rainfall extremes") is a better fit for our manuscript.

Again, we express our gratitude for the suggestions by the three Reviewers and the Associate Editor. And we hope that the new version of our manuscript meets the requirements for publication.

Best regards,

The Authors

**References**

Ashkar, F., Aucoin, F., 2012. Choice between competitive pairs of frequency models for use in hydrology: a review and some new results. Hydrological Sciences Journal 57, 1092–1106. https://doi.org/10.1080/02626667.2012.701746

Hosking, J.R.M., Wallis, J.R., 1988. The effect of intersite dependence on regional flood frequency analysis. Water Resources Research 24, 588–600. https://doi.org/10.1029/WR024i004p00588

Lun, D., Fischer, S., Viglione, A., Blöschl, G., 2023. Significance testing of rank cross-correlations between autocorrelated time series with short-range dependence. Journal of Applied Statistics 50, 2934–2950. https://doi.org/10.1080/02664763.2022.2137115

Serinaldi, F., Kilsby, C.G., Lombardo, F., 2018. Untenable nonstationarity: An assessment of the fitness for purpose of trend tests in hydrology. Advances in Water Resources 111, 132–155. https://doi.org/10.1016/j.advwatres.2017.10.015

Serinaldi, F., 2024. Scientific logic and spatio-temporal dependence in analyzing extreme-precipitation frequency: negligible or neglected? Hydrol. Earth Syst. Sci. 28, 3191–3218. https://doi.org/10.5194/hess-28-3191-2024

Volpi, E., Grimaldi, S., Aghakouchak, A., Castellarin, A., Chebana, F., Papalexiou, S.M., Aksoy, H., Bárdossy, A., Cancelliere, A., Chen, Y., Deidda, R., Haberlandt, U., Eris, E., Fischer, S., Francés, F., Kavetski, D., Rodding Kjeldsen, T., Kochanek, K., Langousis, A., Mediero Orduña, L., Montanari, A., Nerantzaki, S.D., Ouarda, T.B.M.J., Prosdocimi, I., Ragno, E., Rajulapati, C.R., Requena, A.I., Ridolfi, E., Sadegh, M., Schumann, A., Sharma, A., 2024. The legacy of STAHY: milestones, achievements, challenges, and open problems in statistical hydrology. Hydrological Sciences Journal 69, 1913–1949. https://doi.org/10.1080/02626667.2024.2385686

---

## Author Response (AR2)

**Final reply letter - Manuscript 2024-3261**

Dear Associate Editor,

We want to express our sincere appreciation for the attention that you have dedicated to our study, and your prompt response to the submission of our revised version. Also, we are extremely glad that you appreciated and positively evaluated the additional analyses we performed.

We report our replies to your minor comments below.

**Associate editor, prof. Nadav Peleg**

***Associate Editor:***

*Dear Andrea Magnini,*

*Thank you for providing the revised version of the manuscript and your detailed replies to the reviewers' comments. The reviewers are satisfied with your responses and the changes made to the text. However, upon reading the paper, I still have some minor comments that I would like you to address. I am confident that these revisions will not require extensive time and effort. Please find them listed below. Please note that the revised manuscript will be evaluated by me and will not be sent for further peer review. I look forward to receiving the revised version.*

*Sincerely,*
*Nadav Peleg*

*Minor comments:*

*1. In lines 50–54, you list the teleconnection indices and mention that they have a "proven influence on the rainfall regime in the study area". Could you please provide further details here (or in Section 3.1)? Specifically, not about the indices themselves, but regarding their relevance to the study area and whether they have also been used in the context of rainfall extremes in this region in previous studies.*

> **Reply of the Authors:** Many thanks for this suggestion, we have added more details about the teleconnection's relevance in Section 3.1 in the revised article.

***Associate Editor:*** *2. You use the notation "t" to mark tiles, indicating a spatial context. However, "t" is often used to denote time, and since you are also using "st" in the context of time, this might cause confusion. I suggest replacing "t" with "g" (for grid) or another notation that does not imply a temporal meaning and is not already in use in the manuscript. While not critical, this change could improve clarity throughout the text and equations.*

> **Reply of the Authors:** Many thanks for this suggestion, this has been addressed in the revised article.

***Associate Editor:*** *3. Please change the reference to the appendices in the text to "Supporting Material" (e.g., "Supporting Material A" instead of "Appendix A"). Additionally, the supporting material should be presented as a separate file and not included in the main text.*

> **Reply of the Authors:** Many thanks for raising this point, this has been addressed in the revised article.

***Associate Editor:*** *4. Lines 195–198: I would argue that one should select the tile distance in such a way that it would guarantee preserving the local climatology. Alternatively, users may consider not strictly following orthogonal grids, but defining analysis areas based on climatological similarity in rainfall extremes.*

> **Reply of the Authors:** We agree on this point, we inserted this consideration in the Discussion Section, where we think it can find a better place.

***Associate Editor:*** *5. Equation 7 and line 208: I am not clear on what is meant by "sign" in this context. Please clarify.*

> **Reply of the Authors:** Many thanks for raising this unclear aspect. In turn, by "sign" we mean the sign function (i.e., sign(-0.5) =-1; sign(0.5)=+1) on the +/- of a number. This has been clarified in the revised article.

***Associate Editor:*** *6. Figure 3: I recommend using white for the zero-correlation value, with positive and negative values ranging from zero to ±4, reaching dark blue and dark red, respectively. The current colour palette is not very clear. The same suggestion applies to Figure 4.*

> **Reply of the Authors:** Many thanks for this piece of advice, this has been addressed in the revised article.

***Associate Editor:*** *7. I strongly suggest adding a paragraph at the end of the manuscript regarding "Code availability" and sharing the codes used for the analysis, including an example demonstrating its application to one of the stations in your study area. This will enable users to apply your methodology easily and align with FAIR data principles.*

> **Reply of the Authors:** Many thanks for raising this point. We have uploaded a code that performs hierarchical RFA with both the stationary and the doubly-stochastic framework with 13 example stations. This dataset is publicly available in Zenodo (10.5281/zenodo.16610039). Since we do not have the right to publish the real data observed, annual maxima have been slightly altered. This is reported in the new section "code availability".

***Associate Editor:*** *8. Currently, you mention the potential implications of your findings in the introduction and conclusions only, briefly hinting at potential future implementation in the context of non-stationary extreme rainfall analysis. It would be beneficial to include a more detailed discussion (in the discussion section) regarding the implications and potential uses of your findings for improving or reducing uncertainties in extreme rainfall analyses.*

> **Reply of the Authors:** Thank you for this suggestion. A mention of this important topic has been included in the discussion section.

Again, we express our gratitude for the suggestions and the assistance received. We hope that the new version of our manuscript meets the requirements for publication.

Best regards,

The Authors

---

## Author Response (AR3)

**Final reply letter - Manuscript 2024-3261**

**Associate editor, prof. Nadav Peleg**

***Associate Editor:***

*Dear Andrea Magnini,*

*Thank you for uploading the revised manuscript for consideration in HESS. I am happy to accept the paper for publication.*
*Congrats and thank you for contributing to the journal.*

*Sincerely,*
*Nadav*

**The Authors**

Dear Associate Editor,

Again, we want to express our sincere appreciation for the time and attention that you have dedicated to our study. We are grateful for the comments received, which we believe significantly contributed to the improvement of our manuscript.

We are extremely glad that your opinion on the publication is positive, and it is our great honor to contribute to the Journal.

Best regards,

The Authors